# Pleistocene terrestrial warming trend in East Asia linked to Antarctic ice sheets growth

Huanye Wang [1] ✉, Weiguo Liu[1,2], Zhonghui Liu [1,3] ✉, Xiaoke Qiang [1], Xinwen Xu[4], Jing Lei[1,5], Zhengguo Shi[1,6], Yunning Cao[1], Jing Hu[1], Fengyan Lu[1], Hongxuan Lu [1], Xiaolin Ma[1], Youbin Sun [1], Zhangdong Jin [1,6], Hong Ao [1], Zeke Zhang [1], Hu Liu[1,7], Yong Hu[1,8], Hong Yan [1,6], Weijian Zhou [1,9,10] & Zhisheng An [1,6,8] ✉

How terrestrial mean annual temperature (MAT) evolved throughout the past 2 million years (Myr) remains elusive, limiting our understanding of the patterns, mechanisms, and impacts of past temperature changes. Here we report a ~2-Myr terrestrial MAT record based on fossil microbial lipids preserved in the Heqing paleolake, East Asia. The increased amplitude and periodicity shift of glacial-interglacial changes in our record align with those in sea surface temperature (SST) records. However, its long-term warming trend (1.0 °C/Myr, 95% CI = 0.4–1.7 °C/Myr) during 1.8–0.6 Myr ago diverges from the contemporaneous SST cooling. We propose that the Pleistocene warming in East Asia primarily resulted from regionally enhanced heat input and greenhouse effect of rising water vapor driven by Antarctic ice sheets (AIS) growth, highlighting the important climatic effect of AIS evolution. Such long-term warming across the Mid-Pleistocene Transition might have been beneficial for archaic humans' flourishing in Eurasia.

The Pleistocene, about 2.6 million years (Myr) to 11,700 years ago, is an important geological epoch that witnessed the evolution and expansion of our genus *Homo*[1–4]. During this epoch, the cyclicity of Earth's climate shifted from 41 thousand years (kyr) to 100 kyr with an increasing glacial-interglacial contrast, as inferred from the $\delta^{18}O$ record of benthic foraminifera, which reflects the combined signal of deep-sea temperature and land-ice volume or sea level[5]. In the marine realm, paleo temperature reconstructions based on alkenones, Mg/Ca, and faunal proxies in marine sediments indicate that sea surface temperature (SST) generally decreased with a similar rhythm to the benthic $\delta^{18}O$ record[6–8]. In the terrestrial realm, however, due to the

scarcity of suitable proxies and high-quality archives, the long-term Pleistocene temperature history remains insufficiently understood. This limits our understanding of the pattern and mechanism of global climate change during this critical geological epoch, as well as the effects of past temperature changes on the evolutionary trajectory of the human species.

Branched glycerol dialkyl glycerol tetraethers (brGDGTs), a suite of membrane-spanning lipids synthesized by heterotrophic bacteria, are emerging as a popular tool for quantifying past terrestrial temperatures[9–12]. This paleothermometer, developed and well-calibrated with natural climate gradients[13,14], has been validated by

[1]State Key Laboratory of Loess Science, Institute of Earth Environment, Chinese Academy of Sciences, Xi'an, China. [2]University of Chinese Academy of Sciences, Beijing, China. [3]Department of Earth & Planetary Sciences, The University of Hong Kong, Hong Kong, China. [4]Shaanxi Key Laboratory of Earth Surface System and Environmental Carrying Capacity, College of Urban and Environmental Science, Northwest University, Xi'an, China. [5]Xi'an Institute for Innovative Earth Environment Research, Xi'an, China. [6]Institute of Global Environmental Change, Xi'an Jiaotong University, Xi'an, China. [7]School of Water and Environment, Chang'an University, Xi'an, China. [8]National Observation and Research Station of Earth Critical Zone on the Loess Plateau, Xi'an, China. [9]Guanzhong Plain Ecological Environment Change and Comprehensive Treatment National Observation and Research Station, Xi'an, China. [10]Inter-disciplinary Research Center of Earth Science Frontier, Beijing Normal University, Beijing, China. ✉e-mail: wanghy@ieecas.cn; zhliu@hku.hk; anzs@loess.llqg.ac.cn

microbial cultivation[15,16] and molecular dynamics simulations[17] to reflect physiological adaptation to temperature variations. Recently, the application of the brGDGT paleothermometer in a 3-Myr loess-paleosol sequence on the Chinese Loess Plateau (CLP) revealed an unexpected Pleistocene land warming[18], diverging from most marine temperature records. This implies that global temperatures cannot be represented only by marine archives, and the incorporation of terrestrial temperatures is needed for assessing global temperature changes and trends. However, brGDGTs are argued to be likely affected by vegetation coverage and seasonality in soils at the mid-latitude CLP[10], especially during glacial periods. Therefore, how glacial terrestrial temperature evolved during the Pleistocene remains largely unknown[18]. Hence, new terrestrial temperature records are still required to further decipher the characteristics of mean annual air temperature (MAT) evolution during the Pleistocene.

Here, we provide insights into Pleistocene terrestrial temperature evolution by analyzing brGDGTs from a well-dated lacustrine sediment core retrieved from the Heqing Basin (HB)[19] (HQ, 26°34′ N, 100°10′ E, 2190 m a.s.l., 97% recovery) in southwestern China (Fig. 1, see "Methods" section). Our HQ MAT record generally resembles marine SST records at glacial-interglacial timescales; however, its long-term warming trend from 1.8 Ma to 0.6 Ma indicates a decoupling between terrestrial and marine temperatures. This long (~2-Myr), continuous, and highly resolved terrestrial MAT timeseries might provide a benchmark for Pleistocene terrestrial temperature variations from the Asian monsoon region that accommodates more than a fifth of the world's population.

## Results and discussion
### Assessing non-thermal influences on the brGDGT paleothermometer
Despite the great potential of brGDGTs for paleotemperature reconstructions, the impacts of non-thermal factors, such as soil input, changes in water-column structure, water chemistry, and bacterial communities, and seasonal bias should be carefully evaluated prior to the quantitative application of this paleothermometer.

For the HQ core sediments, the influence of soil input on brGDGT-based temperature reconstruction appears to be negligible. In surrounding soils (Supplementary Fig. 1), brGDGTs are characterized by a higher degree of methylation (MBT) and a lower degree of cyclization (DC) than those in lake sediments in the Heqing paleolake (Supplementary Fig. 2a, b). Therefore, while potential soil brGDGT input can bias MBT and reconstructed temperature towards higher values, DC should be biased towards lower values. In our core, however, DC is weakly but positively correlated with MBT ($r = 0.14$, $p = 0.01$) and not correlated with reconstructed MAT ($r = 0.06$, $p = 0.22$) (Supplementary Fig. 3a, b). The ratio of hexamethylated to pentamethylated brGDGT (IIIa/IIa) can also be used to distinguish brGDGT provenance, and a value > 0.92 can generally indicate the aquatic origin of brGDGTs[20]. IIIa/IIa is > 0.92 for 97% samples in the HQ core (avg. $1.56 \pm 0.38$, $N = 380$), much higher than that in surrounding soils (avg. $0.10 \pm 0.05$, $N = 12$) (Supplementary Fig. 2c). Therefore, brGDGTs in the Heqing paleolake should be predominantly produced within the lake, and variations in brGDGT distributions are almost unaffected by soil input.

Within the Heqing paleolake, the impact of water-column structure, water chemistry, and bacterial community changes on brGDGT-based temperature reconstruction should also be minor. First, as methanogens produce relatively high amounts of GDGT-0, GDGT-0/crenarchaeol (GDGT-0/cren) ratio can be used to reflect the relative volume of the anoxic and oxic portions of the water column[21–24]. For 94% of the samples in the HQ core, the GDGT-0/cren ratio is below 1.5 (Supplementary Fig. 2d). This is lower and more stable than that in the Plio-Pleistocene sediments of Lake El'gygytgyn (LE)[22] and the 250-kyr sediments of Lake Chala[23], indicating a relatively stable lake water-column structure of the paleo Lake Heqing. Second, reconstructed temperature can be biased toward high values in lakes with high salinity (conductivity)[14,25]. The abundant carbonate preserved in the HQ core[26] and relatively high CBT′ values (Supplementary Fig. 2e) indicate an alkaline paleolake. However, the occurrence of freshwater diatoms throughout the HQ core[27] and negligible late-eluting isomers after 5-methyl and 6-methyl brGDGTs (which are generally high in brackish and saline lakes[25]) suggest a freshwater environment. Actually, in the global brGDGT dataset for freshwater lakes, while most lakes are

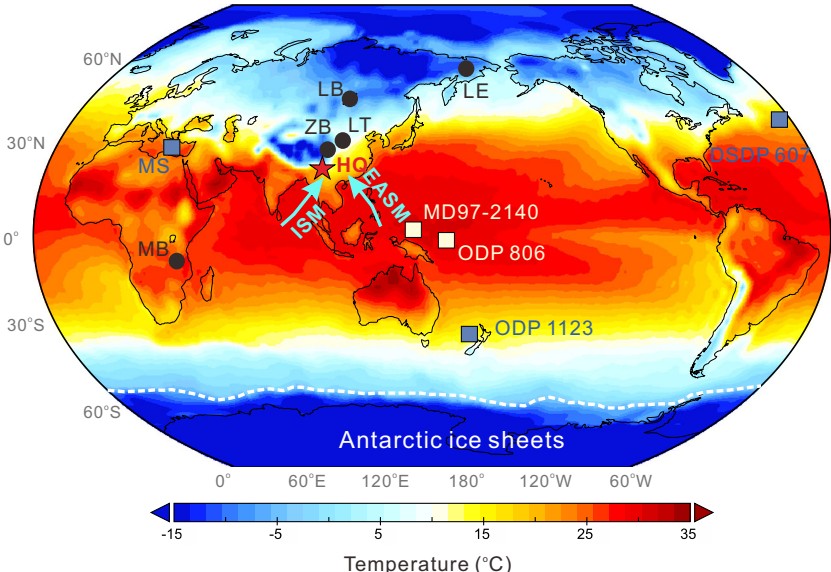

**Fig. 1 | Site locations of Pleistocene temperature records superimposed on the mean annual surface temperature field.** Red star: HB. Black dots: other terrestrial sites including Malawi Basin (MB), Zoige Basin (ZB), Lingtai loess (LT), Lake Baikal (LB), and Lake El'gygytgyn (LE). Light blue squares: deep-sea sites including Ocean Drilling Program (ODP) 1123 (Pacific), Mediterranean Sea (MS), and Deep Sea Drilling Project (DSDP) Site 607 (Atlantic). Light yellow squares: warm pool sites including ODP 806 and MD97-2140. Bright blue lines with arrows: Indian summer monsoon (ISM) and East Asian summer monsoon (EASM). White dotted line: modern Southern Ocean sea ice extent in winter. Climatology data from the NCEP/NCAR Reanalysis were used to produce the basemap (Data provided by the NOAA Physical Sciences Laboratory, Boulder, Colorado, USA, from their website at https://psl.noaa.gov/).

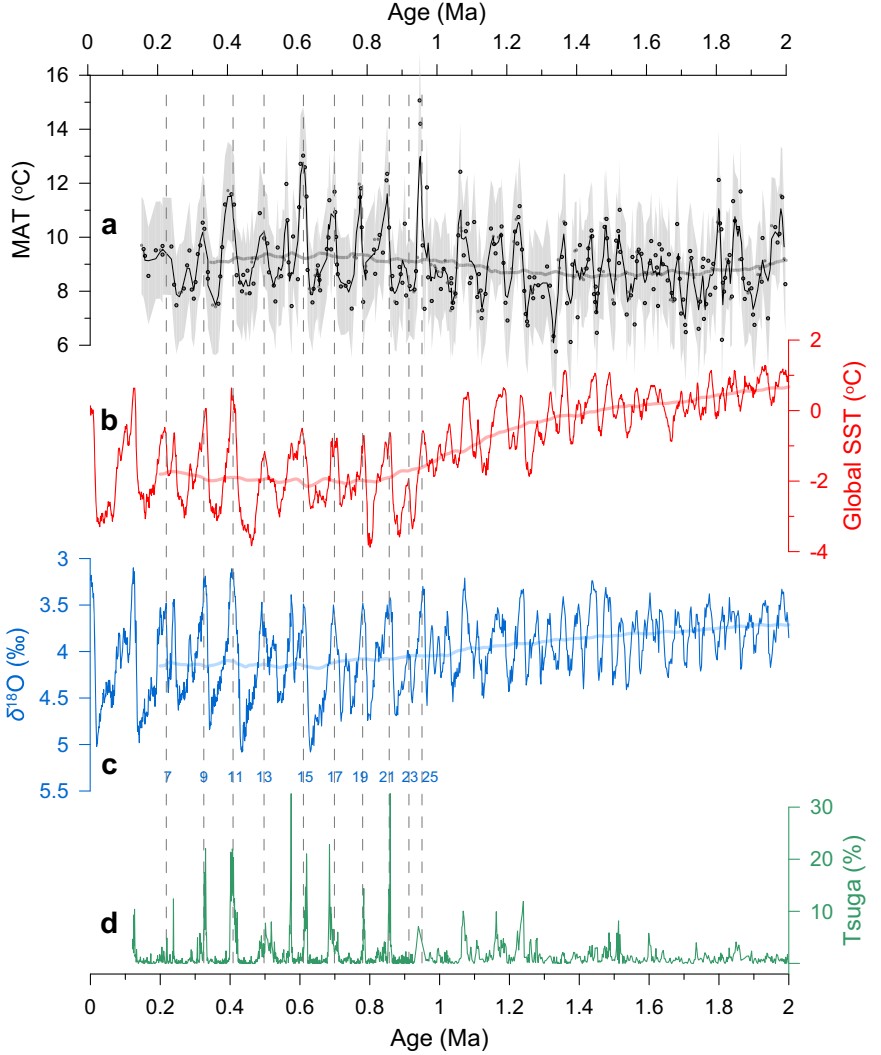

**Fig. 2 | Temperature changes at the HB and in the ocean during the past 2 Myr.** **a** Mean annual temperature (MAT) inferred from fossil microbial lipids at the HB. The gray shading shows the uncertainty of the temperature calibration (±1.8 °C), and the thin line represents a 3-point moving average. **b** The global SST stack[8]. **c** The benthic foraminiferal $\delta^{18}O$ stack[5], controlled by both deep-sea temperature and ice volume changes. **d** *Tsuga* pollen content at the HB[19]. Thick lines are 400-kyr running averages. Vertical dotted lines indicate odd marine isotope stages during 1.0–0.2 Ma.

alkaline, brGDGTs can faithfully track temperature, and there is no significant offset in reconstructed temperatures between lakes with pH = 7–8 and >8 (Supplementary Fig. 4). Furthermore, the effect of brGDGT isomerization ($IR_{6ME}$), reflecting changes in bacterial community related to multiple non-thermal parameters, must be considered when assessing brGDGT methylation as a temperature proxy[28]. $IR_{6ME}$ ranges from 0.33 to 0.75 (Supplementary Fig. 2f), and it is slightly correlated with $MBT'_{5ME}$ ($r = 0.33$, $p < 0.01$) in the HQ core (Supplementary Fig. 3c), implying a likely impact of changes in bacterial community and $IR_{6ME}$. However, the variations of MBT, $MBT'_{5ME}$ and $MBT'_{6ME}$ are quite consistent with each other in our core, particularly after 1.5 Myr ago (Ma) (Supplementary Fig. 5). This contradicts the opposite impacts of $IR_{6ME}$ on $MBT'_{5ME}$ and MBT (or $MBT'_{6ME}$)[29], suggesting that they reflect a common temperature signal, rather than the isomer effect.

Finally, a substantial influence of seasonality on the brGDGT paleothermometer can be excluded for the Heqing paleolake. Given the low latitude setting, the Heqing region experiences restricted monthly changes in air temperature (Supplementary Fig. 1), and when further considering the buffering effect of lake water, seasonal temperature changes could be even smaller. More importantly, it is generally believed that brGDGT-producing bacteria is active above

freezing and therefore brGDGTs can record mean temperature above freezing (MAF) or mean lake water temperature (MLWT)[11,14,29–31]. At Heqing, the relatively high winter air temperature (7.5 °C) suggests that lakes in this region are likely ice-free year-round, and consequently, MAF or MLWT equals MAT. Two lines of evidence further indicate that the growth and preservation of these bacteria/lipids may not depend on seasonal temperature variations in non-freezing lakes: (i) in equatorial lakes spanning a MAT range of ca. 2–25 °C, no correlation between brGDGT concentration (possibly reflecting its production) and temperature is observed[32], and (ii) in the global lake dataset, $MBT'_{5ME}$ correlates more strongly with MAF or MLWT than with mean summer temperature (MST), and additionally, reconstructed growth temperature approximates MAF or MLWT rather than MST (Supplementary Fig. 6).

Overall, brGDGTs in the Heqing paleolake appear to be ideal for tracing MAT variations due to the weak influences of non-thermal factors. The applications of different lacustrine calibrations based on various statistical methods[11,14,29–31] show similar trends and amplitudes of MAT reconstructions, although the absolute values might differ (Supplementary Fig. 7). Nevertheless, we note that reconstructed MAT based on 4 calibrations[11,14,30,31] is weakly but significantly correlated with $IR_{6ME}$ ($p \leq 0.05$) (Supplementary Fig. 3e–h), pointing to a

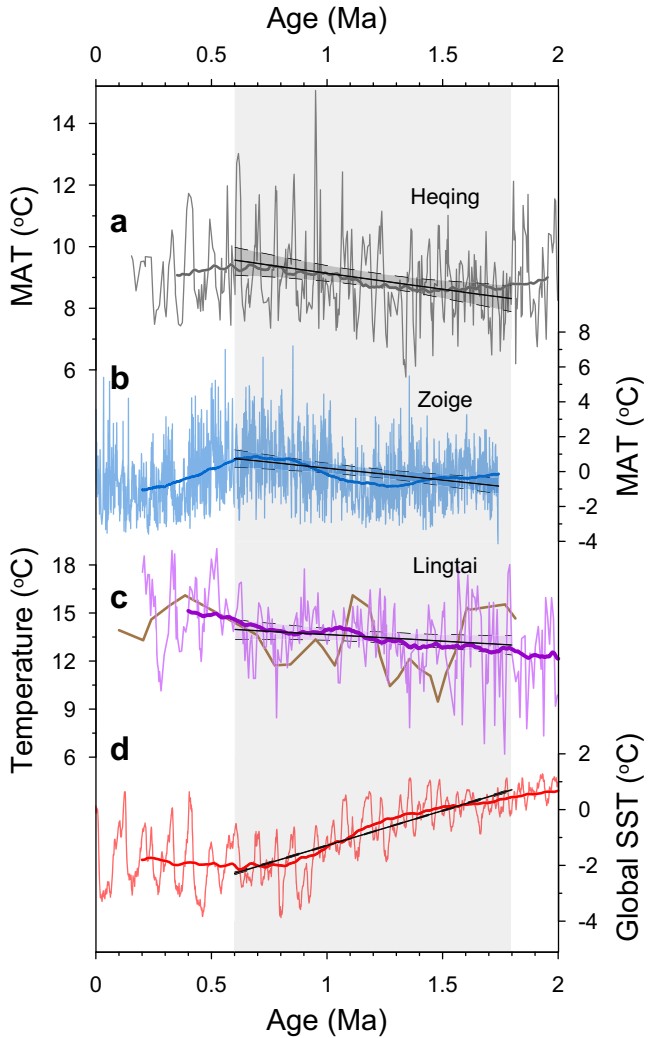

**Fig. 3 | Divergent evolution of terrestrial temperatures and SSTs at 1.8–0.6 Ma.**
**a** MAT inferred from brGDGTs at the HB. **b** Pollen-based MAT records from the ZB, applying the weighted-average partial least squares (WAPLS) approach and the full pollen training dataset[35]. **c** Clumped isotope[36] (brown, 3-points moving average) and brGDGT[18] (purple) -based palaeotemperature reconstructions from LT. **d** The global SST stack[8]. The thick smoothing lines represent 400-kyr running averages, and the straight black lines are the linear regressions for each record (1.8–0.6 Ma) with a 95% confidence interval. The shading highlights the period from 1.8 to 0.6 Ma.

potentially minor impact of brGDGT isomerization on these calibrations. On the other hand, reconstructed temperature based on the recent calibration[29], which may mitigate the isomer effect, has no correlation with $IR_{6ME}$ ($r = 0.08$, $p = 0.14$) throughout the core (Supplementary Fig. 3d), and therefore, this calibration was applied for quantitative temperature reconstruction. Furthermore, to rigorously constrain the potential effects of soil input and water-column structure changes, we excluded samples with IIIa/IIa < 0.92[20] or GDGT-0/cren > 1.5[22,23] for the HQ core. A total of 23 samples were rejected, accounting for only 6% of the 380 samples (Supplementary Fig. 2i). The excluded samples are mainly from the top section of the core, in agreement with the substantial changes in lake depositional environment since ~0.15 Ma inferred from magnetic susceptibility[33] (Supplementary Fig. 2j) and other proxies[19,27].

## Characteristics of Pleistocene terrestrial temperature variations

Our screened biomarker-based reconstruction, with an average resolution of ~5 kyr, shows that MAT at the HB varied from 5.8 °C to 15.0 °C

during the past 2 Myr, mostly (93%) between 7.0 °C and 12.0 °C (Fig. 2a). On orbital timescales, the 100-kyr glacial-interglacial cycles gradually replaced the 41-kyr cycles, with strong periods of both 41 kyr and ~80–120 kyr during the Mid-Pleistocene Transition (MPT, 1.25 Ma to 0.7 Ma) (Supplementary Fig. 8). The average temperature amplitude increased from about 3 °C during the early Pleistocene to 4 °C at 1.0–0.2 Ma (Fig. 2a). Such orbital-scale variations of low-latitude terrestrial temperature, including both cyclicity and amplitude, closely resemble those in the global SST stack[8], tropical SST stack[6], and benthic $\delta^{18}O$[5] (Fig. 2 and Supplementary Fig. 8). This coherent rhythm of terrestrial and marine temperatures indicates that they have followed a common control on orbital timescales during the Pleistocene epoch and confirms the reliability of our brGDGT-based quantitative temperature reconstruction at the HB. The emergence of the 100-kyr periodicity during the MPT, and its full establishment at ~0.6 Ma, possibly reflects an important role of high-latitude ice sheet influence[34] on low-latitude terrestrial temperature variability since the MPT.

A notable feature of the long-term Heqing paleotemperature trend is that, both glacial and particularly interglacial temperatures became warmer from ~1.8 Ma to 0.6 Ma (Figs. 2a and 3a). Such a long-term land surface warming is at odds with the pronounced cooling in marine surface temperatures during this period (−2.5 °C/Myr for global SST[8] and −0.8 °C/Myr for tropical SST[6]). A Mann–Kendall trend test ($S = 4148$, $Z = 3.16$, $p < 0.01$) suggests that the warming trend at the HB was statistically significant. This trend still exists when considering the analytical and calibration error of the brGDGT paleothermometer, as 1000 times of Monte Carlo simulations incorporating the 1.8 °C uncertainty all show warming trends, with an average value of 1.0 ± 0.3 °C/Myr (95% confidence interval = 0.4–1.7) (Fig. 3a, Supplementary Fig. 9), and the proportion of significant ($p < 0.05$) regressions is 78%. Moreover, various lacustrine calibrations[11,14,29–31] yield similar warming trends ranging between 0.7 °C/Myr and 1.6 °C/Myr from 1.8 Ma to 0.6 Ma, and the average value (1.1 °C/Myr) is almost identical to that inferred by the calibration used in this study (1.0 °C/Myr). The above lines of evidence indicate that the long-term warming trend at the HB is statistically robust.

Given the close match of our biomarker-inferred MAT to marine SST records on orbital timescales (Fig. 2) and the minor impacts of non-thermal factors on the brGDGT proxy assessed above, the identified early to mid-Pleistocene terrestrial warming is unlikely due to the defect of the brGDGT paleothermometer. The Pleistocene *Tsuga* pollen record from the same core[19] also lends strong support to our reconstructed temperature record based on microbial membrane lipids. Modern investigations show that *Tsuga* distribution in the Asian monsoon region is constrained by winter temperature at regions with sufficient precipitation[19]. Furthermore, MAT at the HB strongly depends on winter temperature (Supplementary Fig. 10). Therefore, *Tsuga* pollen can be used as a sensitive, albeit qualitative, temperature proxy in this region. The long-term trend in *Tsuga* content and its glacial-interglacial variation both align well with those in our reconstructed MAT (Fig. 2a, d and Supplementary Fig. 11). This further reinforces the robustness of the brGDGT-based quantitative temperature estimates.

The absence of a long-term land surface cooling during the past 2 Myr does not seem to be just a local phenomenon limited to the HB. A 1.74-Myr pollen-based MAT record from the Zoige Basin (ZB), eastern Tibetan Plateau[35] (33° N) indicates a 1.4 °C/Myr warming trend ($p < 0.01$; 95% confidence interval = 0.6–2.1) before 0.6 Ma (Fig. 3b), using all modern pollen data from China and Mongolia as the training dataset. On the CLP, palaeotemperature reconstructions from eolian deposits at Lingtai (35° N) suggest a 0.8 °C/Myr warming ($p < 0.05$; 95% confidence interval = 0.1–1.8) during this period based on the brGDGT paleothermometer[18], corroborated by a low-resolution carbonate clumped isotope record[36] (Fig. 3c). Another low-resolution brGDGT-based paleotemperature record from the North China Plain[37] indicates

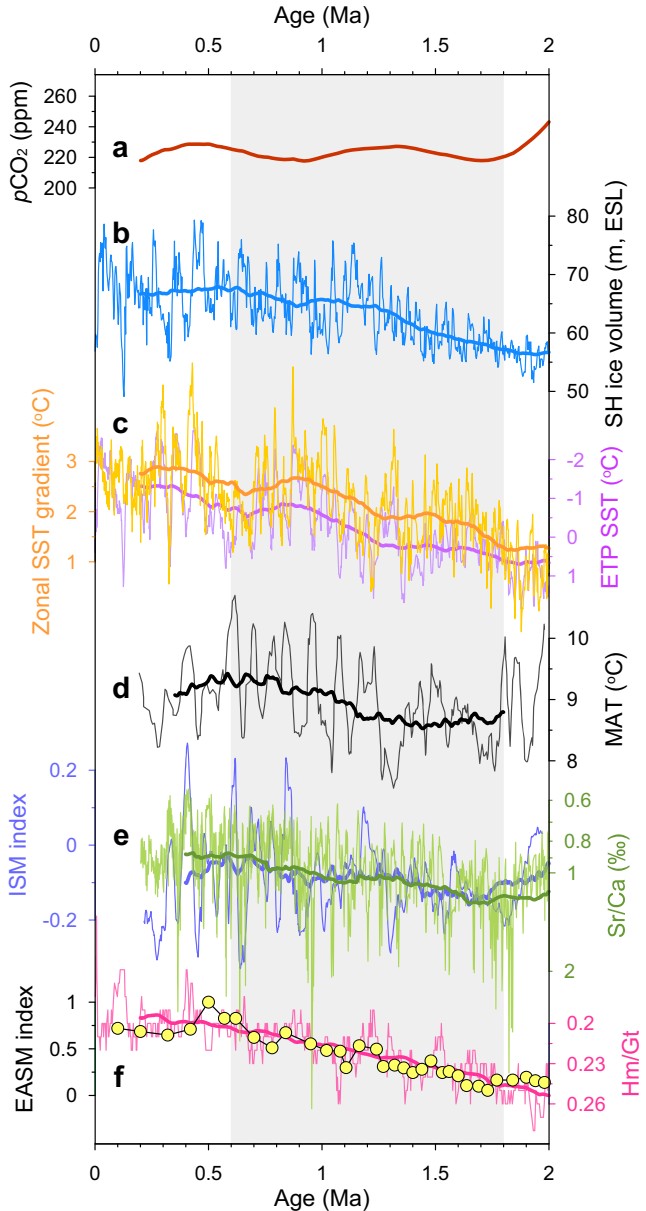

**Fig. 4 | Possible controls on long-term terrestrial temperature variations.**
**a** Pleistocene CO₂ stack of 11 vetted CO₂ records (details in Supplementary Methods). **b** Southern Hemisphere (SH) ice sheet development, in meters of ice-volume equivalent sea-level (ESL)[49]. **c** Eastern tropical Pacific (ETP) SST stack and zonal SST gradient (Indo-Pacific Warm Pool SST minus ETP SST)[8]. **d** Heqing MAT (9-point moving average). **e** ISM index (45-points moving average) and Sr/Ca ratios (a salinity proxy with lower values corresponding to lower salinity, thus higher lake levels) at the HB[19,27]. **f** Hematite/goethite ratio (Hm/Gt) from LT with lower values indicating stronger East Asia summer monsoon (EASM) intensity[53], and EASM rainfall index[51] on the CLP. The thick lines are 400-kyr running averages. The shading highlights the period from 1.8 Ma to 0.6 Ma.

(such as East Asia), warm pool and deep sea, might have diverged from the general SST trend during the past 2 Myr. The divergent Pleistocene temperature trends, which have not attracted sufficient attention from the paleoclimate community previously, indicate that the Earth's climate system has possibly evolved into a more complex mode since ~2 Ma.

## Possible controls on long-term temperature variations in East Asia

The long-term warming trend during 1.8–0.6 Ma observed for the 3 East Asian records at Heqing, Zoige, and Lingtai, in contrast to the contemporaneous global sea surface cooling, cannot be primarily attributed to the relatively high elevation of these terrestrial records. Observational data and numerical models for global temperature variations suggest that, elevated land surfaces warm faster than those near sea-level during the past several decades[46]. During the last glacial maximum (LGM), GDGT-based temperature reconstructions from East African lakes show amplified cooling with elevation, resulting in a significantly steeper lapse rate than today[47]. Both modern and LGM data show that at higher elevations, temperature amplification is greater than near sea level, but the warming or cooling trends at different elevations cannot be inverted. Based on ERA5 reanalysis, HadCRUT5 dataset, and CMIP6 historical simulations (1959–2014), land surface warming is on average 16% higher at 2190 m a.s.l. than that at sea level across the tropics and subtropics (40° S to 40° N)[46]. Assuming a similar elevation-dependent warming at Heqing, the warming trend should be 0.9 °C/Myr at sea level, still opposite to the 2.5 °C/Myr cooling for global SST.

Atmospheric carbon dioxide (CO₂) is considered the main greenhouse gas responsible for current global warming and the primary forcing for past temperature changes[48]. Based on the recent compilation of vetted and modernized data from various proxies[48], no long-term trend in CO₂ level is observed from 1.8 Ma to 0.6 Ma (Supplementary Fig. 13a). One might argue that this synthesis might mostly rely on records with higher density of datapoints, as its variation resembles that for the record inferred from δ¹³C of terrestrial C₃ plant remains, which accounts for 30% of the total datapoints during 2.6–0.2 Ma (Supplementary Fig. 13a). To reconcile the potential artifact caused by this issue, as well as the systematic bias and large scattering of each work using different methods and archives, we further compiled a CO₂ stack by normalizing 11 long Pleistocene CO₂ reconstructions vetted by the CenCO2PIP Consortium[48] (Supplementary Methods). This stack still indicates no long-term trend in CO₂ level during the past 2 Myr (Fig. 4a). Such a relatively stable CO₂ level can partially explain the absence of a long-term Pleistocene cooling on land, but is insufficient to drive the long-term terrestrial warming at East Asia (Fig. 3a–c).

We hypothesize that the long-term terrestrial warming trend from 1.8 Ma to 0.6 Ma in East Asia might be dynamically linked to Antarctic ice sheets (AIS) growth and its associated feedbacks. The growth of AIS at 2.0–0.6 Ma (Fig. 4b) could have resulted in substantial cooling and sea ice expansion in southern high latitudes[49]. The high-latitude SST cooling and its equatorward propagation through the Peru Coastal Current or the atmosphere and its thermal coupling within the ocean[50] can then lead to pronounced SST decrease in the equatorial eastern Pacific, thus strengthening the tropical zonal SST gradient and Walker circulation (Fig. 4c). Enhanced Walker circulation (or the La Niña-like condition) can both promote relatively increased heat accumulation in the western Pacific (Supplementary Fig. 12c) under the background of global sea surface cooling (Fig. 3d), and strengthen Asian summer monsoons[51,52]. Changes in the cross-equatorial pressure gradient due to the asynchronous development of bipolar ice sheets can also strengthen Asian summer monsoon circulations[49]. The progressive strengthening of monsoons has been widely documented at the HB and CLP[19,27,51,53] (Fig. 4e, f). As the Asian summer monsoon can transport

no obvious long-term trend during the Pleistocene, despite the complex changes in sedimentary facies. Beyond East Asia, available long Pleistocene terrestrial temperature records also show no obvious cooling trend from 1.8 Ma to 0.6 Ma or during the MPT; however, these records are generally fragmentary or argued to be affected by non-thermal factors (Supplementary Discussion). Moreover, in the marine realm, temperatures during 1.8–0.6 Ma were also relatively stable or slightly increased in the deep sea[38–40] (except for the Atlantic record[41]) and the western Pacific warm pool[8,42–45] (Supplementary Fig. 12). Therefore, long-term temperature trends at some terrestrial regions

warm and humid air masses (including water vapor) from tropical oceans (including the warm pool with increased heat accumulation) to the Asian continent[54], an intensified summer monsoon can thus warm the continent through sensible heat flux and latent heat release[55]. Moreover, the rising water vapor could also amplify regional warming[56,57] as it is an important greenhouse gas in the atmosphere. Our numerical simulation (Supplementary Methods) also shows that the expansion of AIS can potentially increase tropical zonal SST gradient and elevate surface temperature over much of the Eurasian continent, with an overall strengthening in the Asian summer monsoon (Supplementary Fig. 14), despite that it might be simplistic and may not exactly capture the real processes of AIS expansion.

In summary, during 1.8–0.6 Ma, both glacial and interglacial temperatures might have increased on land (particularly East Asia). This is likely driven by a series of processes caused by the AIS growth under a relatively long-term stable global $CO_2$ level, highlighting the importance of AIS evolution on global climate change. Our quantitative MAT record challenges the recent climate model simulations[3,58] which yield a decreasing trend in global terrestrial temperature during the past 2 Myr, forced with a modeled gradual lowering of atmospheric $CO_2$[59] that differs from proxy-based paleo $CO_2$ reconstructions (Supplementary Fig. 13). Based on marine temperature reconstructions[5–8] and climate model simulations[3], it was believed previously that the global climate cooled during the Pleistocene and therefore the dispersal of hominins into extratropical regions during this period[1,4,58,60] was thought to be related to improved adaptability of archaic humans to cold environments[2,60]. Based on our Heqing and some other terrestrial records (Figs. 3 and 4), however, the reconstructed long-term warming and wetting trends on land (at least Asian monsoon regions) imply that extratropical Eurasia might have gradually become more suitable for the survival of hominins from the early to mid-Pleistocene. This should have been directly beneficial for our ancestors to flourish in Eurasia across the MPT[1,3,4,58,60], not necessarily requiring that they could adapt to climate stress such as extremely cold and arid environments. Particularly, the slightly increased glacial temperature might have facilitated hominins to survive through the prolonged glacial times, although cultural innovations might have also played an important role[60]. Moreover, our results highlight the possible divergent evolution of land surface temperatures (at least in some regions) and marine temperatures, and the processes involved could be helpful for projecting future temperature changes with regional patterns identified.

## Methods

### Study site, sampling, and chronology
The Heqing paleo-lake is located at the center of the HB in Yunnan Province, southwest China. Pilot geophysical surveys indicate that ancient sediments accumulated up to 700 m in the basin since the late Cenozoic, and therefore the basin might serve as a potential terrestrial archive in the Indian monsoon region[61]. At the HB, the mean annual, January, and July temperatures are approximately 13.8 °C, 6.8 °C, and 19.2 °C, respectively (Supplementary Fig. 1). The mean annual precipitation is 962 mm, with over 80% occurring from June to September due to the influence of the Indian summer monsoon (ISM). The regional vegetation is dominated by northern subtropical evergreen forest[19,26].

In 2002, a 665.83-m (calibrated depth) long sedimentary core (HQ) was retrieved from the center of the lake basin (26°33′43″ N, 100°10′14″ E, 2190 m a.s.l.). Internal plastic tubes were used during drilling to avoid twist and distortion of the sediment core[27]. The whole core recovery is higher than 97%[19], allowing for high-resolution and continuous paleoclimatic reconstructions. Laminated greyish-green calcareous clay and silty clay dominate the core, with thin-bedded silt and fine sand layers, except that two intervals of sand layers with fine gravels occur at 372.5–371.9 m and 195.4–189.5 m. Aqueous herb

pollen and freshwater diatoms are found throughout the sequence, suggesting that the sediments were of typical lacustrine origin[19,27,62]. Moreover, 12 surface soils were collected surrounding the Heqing paleolake in 2025 (Supplementary Fig. 1). For each soil sample, three randomly collected subsamples (upper 5 cm) were pooled and mixed to make one composite sample representing that location.

The chronology of HQ drilling core has been well-established by magnetostratigraphy, radiocarbon dating, and astronomical tuning[19]. Briefly, paleomagnetic measurements of the thermal demagnetization were used to generate the geomagnetic polarity sequence of the HQ core, and the framework of the core was then established by correlating it with the geomagnetic polarity time scale (GPTS)[63]. The Matuyama/Gauss (M/G) boundary (-2.6 Ma) is located at 614.47 m, indicating that the age of the 665.83-m core can be extended back to the Pliocene epoch. Combined paleomagnetic and radiocarbon analysis identified that the Laschamp Excursion event (-40 ka[64]) occurs at ~4.5 m. Finally, a refined astronomical time scale was developed by simultaneously tuning the filtered 41- and 21-kyr components of the *Tsuga* pollen content to Earth's orbital obliquity and precession parameters[19].

### GDGT analysis and proxy calculation
Generally, 1–3 g homogenized sediment and soil samples with a known amount of $C_{46}$ internal standard[65] were extracted with dichloromethane (DCM): methanol (9:1) using a Dionex™ ASE™ 350 at 100 °C and 1500 psi. The total lipid extract was then subject to base hydrolysis, and the extracted neutral fraction was eluted by *n*-hexane and MeOH:dichloromethane (1:1, *v/v*) on a silica gel column to separate it into apolar and polar fractions. After filtration, the polar fraction was analyzed on a high-performance liquid chromatograph/atmospheric pressure chemical ionization-mass spectrometer (HPLC/APCI-MS) system (with a Shimadzu LC-MS 8030) at the IEECAS. GDGTs were separated on two coupled silica columns (250 mm × 4.6 mm, 3 μm; GL Sciences Inc.) using isopropanol and *n*-hexane as elutes[18]. Selected ion monitoring (SIM) mode was used to target specific $[M+H]^+$ ions for GDGTs and the internal standard ($C_{46}$), and subsequently GDGTs were quantified by integration of the peak area of $[M+H]^+$ ions, and comparison to that of $C_{46}$.

The MBT, MBT′$_{5ME}$, MBT′$_{6ME}$, DC, CBT′, and IR$_{6ME}$ indices of brGDGTs were calculated as follows[13,66–68]:

$$MBT = \frac{Ia + Ib + Ic}{All\ brGDGTs} \quad (1)$$

$$MBT'_{5ME} = \frac{Ia + Ib + Ic}{IIIa + IIa + IIb + IIc + Ia + Ib + Ic} \quad (2)$$

$$MBT'_{6ME} = \frac{Ia + Ib + Ic}{IIIa' + IIa' + IIb' + IIc' + Ia + Ib + Ic} \quad (3)$$

$$DC = \frac{Ib + 2*Ic + IIb + IIb'}{Ia + Ib + Ic + IIa + IIa' + IIb + IIb'} \quad (4)$$

$$CBT' = \log\left(\frac{Ic + IIa' + IIb' + IIc' + IIIa' + IIIb' + IIIc'}{Ia + IIa + IIIa}\right) \quad (5)$$

$$IR_{6ME} = \frac{IIIa' + IIa' + IIb' + IIc'}{IIIa + IIa + IIb + IIc + IIIa' + IIa' + IIb' + IIc'} \quad (6)$$

Fractional abundances of individual brGDGTs, GDGT proxies, and brGDGT concentration were provided in Figshare (DATA AVAILABILITY). The IR$_{6ME}$ values vary from 0.33 to 0.75 for our brGDGT data, and therefore, we applied the recent calibration[29], which can effectively correct the isomer effect on the brGDGT paleothermometer.

Replicated pretreatment and analysis of 10 samples suggests an average analytical error of 0.3 °C, while the RMSE for the calibration is 1.8 °C. Therefore, we conservatively compounded the calibration error with analytical error such that the total error for our temperature reconstruction = sqrt(calibration error$^2$ + analytical error$^2$)[69] = 1.8 °C. We should point out that the calibration error is possibly caused by the uncertainties in obtaining growth temperature for each sample in the calibration dataset, as well as the global spread of samples and the accompanying large variations in other non-thermal environmental parameters. At a single site and for a specific core, GDGTs can more accurately record the relative temperature changes (such as the temperature offsets or trends) than the absolute values, as it can reduce systematic calibration error and site-specific effects[47,69,70]. Monte Carlo simulations were further performed on Python to calculate the uncertainty for the linear trend of temperature variation during 1.8–0.6 Ma (CODE AVAILABILITY), under the 1.8 °C error of absolute temperature values.

## Data availability

All new data generated in this study have been deposited in Figshare: https://doi.org/10.6084/m9.figshare.29877254.v1. Source data are provided with this paper.

## Code availability

The *Acycle* software is publicly available at: https://acycle.org/. The Python code used for regression and uncertainty analysis with Monte Carlo simulations has been deposited in Figshare: https://doi.org/10.6084/m9.figshare.29323550.

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

## Acknowledgements

We thank the Chinese Environmental Scientific Drilling (CESD) Program for providing samples. This research was financially supported by the National Natural Science Foundation of China (42122021 to H.W.), the Fund of Shandong Province (LSKJ202203300 to Z.A., Z.L., and H.W.), the National Key Research and Development Program of China (2023YFF0804300 to H.W.), the National Natural Science Foundation of China (42173014 and 42273030 to W.L. and H.W.), the Chinese Academy of Sciences (CAS) (XDB40000000 to W.L. and W.Z.), and the Youth Innovation Promotion Association, CAS (2019403 to H.W.).

## Author contributions

Z.A. and W.L. designed the research. H.W., W.L., Z.L., Y.C., J.H., X.Q., X.X., F.L., H. Lu and Y.H. performed data analysis. Z.S., J.L. and Z.Z. conducted model experiments. H.W., Z.L., Z.J., X.M., H.A. and H. Liu collected data. Z.S., J.L., Z.Z., Y.S., H.Y. and W.Z. provided feedback on the analysis, the figures, and the manuscript. H.W., W.L., Z.L. and Z.A. led the writing of the manuscript with intellectual contributions from X.Q.,

X.X., J.L., Z.S., Y.C., J.H., F.L., H. Lu, X.M., Y.S., Z.J., H.A., Z.Z., H. Liu, Y.H., H.Y. and W.Z.

## Competing interests

The authors declare no competing interests.
