## [Transparent Peer Review file · Nature Communications]

Pleistocene terrestrial warming trend in East Asia linked to Antarctic Ice Sheets growth

Corresponding Author: Professor Huanye Wang

Version 0:

Reviewer comments:

Reviewer #1

(Remarks to the Author)

The manuscript by Wang et al. presents new estimates of terrestrial mean annual temperatures (MAT) from a low-latitude site of China covering the last 2 Ma. The results add to the growing list of arguments for a long-term decoupling of regional terrestrial temperatures from benthic temperature trends and changes in ice volume during the Middle Pleistocene. The novel aspects of this study include namely the geographic location of the dataset, which represents the first MAT estimate for the low latitudes of Eurasia. The low latitude setting makes this dataset valuable due to a relatively weak seasonal bias (aliasing), but this part of the manuscript needs a more careful argumentation, in my opinion (please see “specific comments” below).

The manuscript is well written; a few typos and stylistic issues are listed below. I would also suggest improvements to the structure of argumentation (please see below).

Methods are described adequately, in my opinion, and conclusions appear supported by the data, although I am not able to fully evaluate veracity of the GDGT proxy due to my limited expertise. The hypothesized mechanism of the long-term temperature decoupling builds upon previous literature (which is referenced), and is partly supported by paleoclimate model in the supplement.

The results represent an important contribution to our understanding of the climate evolution and high–low latitude interactions during the Pleistocene. I enjoyed reading about the proposed implications for the migration of hominins to Eurasia.

SPECIFIC COMMENTS:

1) brGDGT proxy at Heqing

The authors propose that the low-latitude lacustrine record of brGDGT is not strongly affected by seasonality (line 72), which seems reasonable given the low latitude setting. However, considering the sensitivity of MAT to winter temperatures at the Heqing site (line 105), I would expect some discussion on the origin of brGDGT in the context of seasonal variations in temperature and precipitation. The discussion on lines 146-163 appears incomplete. Please explain: are the GDGT-bearing bacteria autochthonous to the lake or were the lipids transported to lake from soil profiles? What is the role of monsoonal seasonality in the growth and preservation of these bacteria/lipids? I understand the argument with *Tsuga* content on lines 154-157, but considering that this winter proxy varies on glacial/interglacial time scales, is it reasonable to neglect seasonality as one of the controls on the GDGT dataset? Please try to address these questions in the discussion.

2) Structure of argumentation

The section “Possible controls...” (lines 146-..) includes the main discussion of the mechanism of temperature decoupling. The first paragraph, addressing the possible seasonal bias in the GDGT proxy, does not fit here. I would suggest moving this paragraph higher, as a separate section (perhaps “The role of seasonality...” or “Seasonal bias...”?).

3) Language

L20-24: this is a terrible sentence; please rephrase, split in two

L36: "flourish" is a verb; please rephrase "facilitated the flourish"

L46: perhaps not necessary to explain that we inhabit the terrestrial realm.

L90: "full" instead of "fully"

L132-136: sentence too long, hard to understand; please rephrase

L192: "Progressive" instead of "progressively"

L199: "albeit quite qualitatively": what do you mean? please explain

Reviewer #2

(Remarks to the Author)

Wang et al. show a new Pleistocene (2 - 0.2 Ma) terrestrial temperature reconstruction from a paleolake in southwestern China using fossilized microbial lipids (brGDGTs). The study presents a valuable high-resolution temperature record from a region and an archive (land temperatures) where there is little long-term continuous Pleistocene temperature reconstruction. They hypothesized that Antarctic Ice Sheet expansion is the cause/mechanism of a warming during the MPT in terrestrial archives, which they differ from Global SST. However, I don't think that the hypothesized mechanisms presented here, the references of additional studies to support their hypothesis and the extended data support their study/hypothesis. I have a lot of main concerns and I think a re-evaluation of what I think is an over-interpretation of the data needs to be done.

Main comments:

My main concern is the over-interpretation of the temperature change seen in the record. When taking into account the RMSE of the calibration and standard deviation of the data, it's hard to see how this warming can be separated from the natural variability of the reconstructed temperature. A 1°C temperature change from a proxy that has 1.8°C (keep in mind this is +/- 1.8°C) it's a big stretch in interpreting significant temperature changes to claim such mechanisms/conclusions.

Second, I am concerned about the references. First, they are not only wrong (the number doesn't correspond to the reference), but also I could not find the same data or information reported by the references in some cases.

1. Lake Baikal reference number is wrong, and the closest reference of it doesn't show the BioSi reconstructions that are shown here in Figure 3. The paper uses Biogenic Si to identify glacial and interglacial sections but it has not been used as a long-term temperature reconstruction. This is mostly because BioSi is subjected to dissolution, so you can't use the abundance of it to say that there is a temperature trend over time, you use it to identify warmer intervals than others to then do cyclo-stratigraphy or studies like that, not to infer quantitative temperature changes over time.

2. Lake Malawi, the proxy used here is TEX86 which has its own issues in freshwater lakes, but more importantly, the original paper doesn't claim to have an increase in temperature because the variability in the data is within the uncertainty (2sigma = 0.8°C). Additionally, this study clearly showed CO₂ as the main driver of significant temperature changes along the record and the overall conclusion of it had to do more with a wetter/drier climate than temperature changes.

3. Lake El'gygytgyn: this warming rate you mention here 1. Is not significant (see p-value) and the record doesn't show cooling around the MPT (in agreement with your hypothesis) but also it does not show warming. The main change that happens at this high-latitude site is a change in hydrological conditions.

4. In line 435, reference 24, this paper doesn't show anything about the formation and demise of the Heqing paleolake. I could not find such a reference. Also, I don't know why you dismissed those intervals? Even though they don't represent a full lake according to what you said they could give you more information about changes during key periods of time (last interglacial to LGM and Plio-Pleistocene) that have been shown to have a more extreme or a more drastic change that could help differentiate noise vs. signal.

5. Reference 81 and 82. These are not peer-review articles. 81 is a public-facing article from a science journalist and 82 is a conference abstract. I don't think you should cite non-peer review references in a manuscript submitted to Nature Communications.

Third, one of your data that supports your hypothesis is your extended data figure 6 which is the results of the model output of PI-AIS(PI). First, I think the figure should benefit from having labels ("a", "b", etc...) and a more descriptive figure caption. Second, the results in this figure do not clearly support your hypothesis or the figures should be presented differently. I think you could benefit from having a box or an arrow that can show what you mean in the text. For example, you have "shows that the expansion of AISs can potentially increase tropical zonal SST gradient", where in the figure you can see that? I don't see these maps and it is obvious to me to see such an increase in tropical zonal SST. Also, you have "elevate surface temperature over Eurasia" which regions are you talking about because there is a combination of warm and cold regions over Eurasia. You have "with an overall increase in water vapor in the Asian monsoon region", again, highlight the regions you are talking about in these maps because I see some strong red colors over India/Tibet but a blue color around South China where I think your core comes from. Anways, I don't see how these results clearly support your statements. By the way, next to the water vapor panel there is an "x" above 30N, what's that?

Minor comments:

Add elevation in abstract? I think that when talking about temperature reconstruction its good to know if the terrestrial temperatures are near sea level or high elevation sites.

Add modern MAT in the figures to see how it compares with your reconstructed temperatures? The general thought is that the last million years were colder than today.... So I want to see how modern MAT looks with your reconstructed temperature and if your estimate is colder or warmer than that as your present absolute values.

You used the Global SST from Clark, but they also present SST compilations by basin. I think that rather than using individual cores near your location you can add how the SST evolution in the Warm Pool looks in comparison with your record. Then, you can evaluate other mechanisms of why you don't see a clear temperature change over the MPT like other global records.

Line 186-187: you are claiming that there is a decrease in SST in the equatorial eastern Pacific that is strengthening the tropical zonal SST gradient and walker circulation, but when looking at Figure 4c.. 1. I think the records should go all the way back to the beginning of the Pleistocene, and 2. you can have that data with the new Global SST compilation by basin from Clark et al. 2024.. And probably highlight where the "pronounced decrease" happens because with my eyes, in that figure, I don't see any "pronounced" change.

You report a 1°C/myr... but what's the error or standard deviation of this?

Line-by-line

Line 60-61: "This implies that Pleistocene global temperature, or at least terrestrial temperature, cannot be simply represented by marine temperature." I think you should re-arrange this sentence by saying something like global temperatures cannot be represented only by marine archives and the incorporation of terrestrial temperatures is needed for assessing global temperature changes and trends. Or something along the lines that global temperature are both terrestrial and marine, not that terrestrial temperatures are not represented by marine (that is obvious)

Line 62: Do you mean glacial-interglacial? Or just glacial? Is just strange to only mention glacial if you are referring to the entire pleistocene. Also that sentence could be two sentences rather than one.

Line 69: what's the elevation?

Line 80: looking at the SD Fig.1. Seems to me that the gradual transition really starts at ~1.2Ma in your record... not before... no?

Line 89-92: is there a reference to cite here? As it is written it seems like there is a need for a larger evidence of that high-latitude ice-volumen forcing .

Line 197: reference?

Line 704: what is ESL, I don't see it defined anywhere.

Reviewer #3

(Remarks to the Author)

Review of Wang et al. for Nature Communications

Joseph B. Novak

Recommendation

Reject and encourage resubmission.

Summary

Wang et al.'s manuscript entitled "Pleistocene terrestrial warming trend linked to Antarctic Ice Sheets Growth" presents a 2-million-year terrestrial paleotemperature record from Heqing paleolake in southwestern China. The reconstructed paleotemperatures show a warming trend, rather than cooling, over the 1.8–0.6 Ma interval, in contrast to marine sea surface temperature records, which the authors attribute to dynamical changes in the Earth's atmospheric circulation in response to Antarctic Ice Sheet growth.

The lack of a rigorous assessment of the branched GDGT biomarker distributions used to generate the paleotemperature record is seriously concerning. While the authors argue that the similarity between the reconstructed paleotemperatures to a pollen record in the same sediment core indicates that the brGDGTs are primarily recording temperature, I remain unconvinced, especially since this core comes from what appears to have been an alkaline lake based on the occurrence of carbonate sediments (Shen et al., 2007; ref. 52 in this manuscript). The presentation of the pollen data on a log-scale is somewhat odd and, possibly, misleading since pollen data are usually presented on a linear y-axis scale.

A thorough assessment of any possible non-thermal influence on the brGDGT distributions in this lake is fundamental to this study. The lack of this needed supplementary discussion ultimately motivates my recommendation to reject the manuscript. However, the data presented here hold the potential to be a landmark dataset in the field if the authors can show that these

non-thermal factors are not influencing the brGDGT biomarkers here. Therefore, I encourage the resubmission of this manuscript after my comments are addressed.

Major Comments

Extended Data Figure 3: Why is the Tsuga data plotted on a log scale? This is odd to me since pollen data are typically plotted on a linear scale. Is there a justification for this or is it simply to present that data in such a way that they appear to closely resemble the brGDGT MAT reconstruction? Similarly, is the strength of the correlation shown in the scatterplot dependent on the log transformation of the Tsuga pollen data? Is there perhaps some sort of bulk geophysical proxy data (magnetic susceptibility, bulk density) that could be used to make this comparison more effectively?

L146 and Figure 3: The authors make a statement here regarding “the absence of a long-term terrestrial cooling during the past 2 Myr.” However, the data presented in the manuscript only represent a relatively small number of locations and narrow band of latitudes. In Figure 3, the authors show data from five sites. One dataset (Lake Baikal) is not a paleotemperature dataset (see my comment below). The remaining four datasets are from the tropics or near tropics, three of which are from China spanning a very narrow range of latitude (27–35°N). So, essentially, are the authors claiming that Southern China and Eastern Africa are representative of terrestrial temperature change throughout the entire Earth? This is unlikely, considering that modern climate change is resulting in remarkably different rates of temperature change in the high latitudes vs. the tropics. See also the pollen paleotemperature record from Lake El'gygytgyn, which shows a long-term cooling since the Pliocene (see Fig. 3d of Daniels et al., 2021 shows this nicely [<https://doi.org/10.1002/jqs.3347>]). Although these data do not sample every glacial-interglacial cycle, they appear to directly contradict the claim made here and are not discussed in the manuscript.

L457–463: Why is there no supplementary text or figures showing the long-term trends in IR6Me, CBT', brGDGT concentration and GDGT-0/Cren ratio in these samples? It is very important to understand the potential for non-thermal factors to influence brGDGT paleotemperature reconstructions, especially in what the authors consider a “terrestrial MAT time series benchmark for Pleistocene terrestrial temperature variations.”

There are several important factors that must be discussed for the community to understand whether the brGDGT distributions are accurately capturing temperature. These include: (1) brGDGT provenance; (2) the potential for changes in lake water chemistry to affect the brGDGT distributions; (3) whether there were changes in other aspects of the brGDGT distributions that are correlated with the warming inferred between ~1.2–0.2 Ma in Heqing Basin.

For example, the brGDGT paleotemperature reconstruction method used here (developed by the same lead author) is supposed to disentangle the relative contributions of 5-methyl and 6-methyl brGDGT producers when estimating paleotemperatures (i.e., remove the potential temperature bias from multiple brGDGT-producing bacterial communities contributing lipids to sediment). Are the resulting paleotemperature estimates from this method correlated with IR6Me, CBT', BIT, etc. in these samples? If so, does this imply that there is a substantial non-thermal influence on the temperatures reported here that is not accounted for by this method? If not, does that mean the method is working? We do not know because this is not explored.

The overarching point of this comment is that there is not sufficient information provided here for me (or anyone else) to judge whether the brGDGT distributions are authentically capturing temperature in the Heqing paleolake record. This is particularly important to consider since the sediments in this lake core are calcareous clays, suggesting that the lake was alkaline. These alkaline lakes appear to be especially challenging environments for brGDGT paleotemperature reconstruction, for example Lin et al. (2024) [<https://doi.org/10.1016/j.earscirev.2024.104694>]. Indeed, Shen et al. (ref. 52 here) report large changes in sedimentary carbonate content in this lake core. Shen et al. described substantial changes in the amplitude of these sedimentary carbonate cycles through time and the deposition of gravels associated with regional mountain uplift throughout the core. Are these sedimentary features, indicative of changes in lake water chemistry and, perhaps, changes in brGDGT provenance, respectively, related to any aspects of the paleotemperature record presented here?

In my view, this manuscript cannot pass review until the potential non-thermal influence on the brGDGT distributions is addressed. I appreciate the huge amount of effort that went into analyzing this dataset and I strongly encourage the authors to address these comments and resubmit their work. Long terrestrial paleoclimate records are a rare and special thing with the potential to capture the thought and attention of the scientific community. This is why we must treat them carefully, and critically, to be sure about their underlying quality.

Supplementary data: Why is there no supplementary data table with the reconstructed temperatures and underlying brGDGT distributions? It is standard practice to share this information.

Minor Comments

L19: “Our knowledge of Earth’s temperature history” rather than “knowledge on.”

L36: “flourishing” rather than “flourish.”

L42 and elsewhere: The SI convention now is to italicize the δ in $\delta^{18}\text{O}$.

L73: timeseries should be written as one word, not two.

L90: "full establishment" rather than "fully establishment."

L93: Consider rephrasing to: "A notable feature of the long-term Heqing paleotemperature trend is that..."

L96: "trend" in "pronounced cooling trend" can be removed as it is unnecessary.

L104–106: I do not really follow what you mean here. Mean annual temperature by definition incorporates the temperature of all months equally. Is this statement trying to parse the slight difference in the correlation strength between MAT and summer vs. winter temperature? This seems like a bit of a stretch to me since the difference in Pearson's r here is rather minor.

L116–117: does this imply that a different result is obtained if a more restricted pollen training dataset is used?

L124–126: The Lake Baikal biogenic silica record does not track paleotemperature directly and certainly not in a linear fashion. I therefore urge extreme caution in interpreting this record as a paleotemperature proxy dataset as is done here. A major drawback of this data type is that it cannot distinguish relative differences in how cold individual glacial periods were. This is because it is impossible to have less than 0% biogenic silica content, meaning that any potential differences in temperature between glacial periods is missed by the proxy signal. You will notice that this is essentially the value of every glacial period (at least the stadials, interstadials have slightly higher content) over the entire 1.8-million-year period covered by the Prokopenko et al. 2006 record you reference here.

See Prokopenko et al. 2001 [doi:10.1006/qres.2000.2212] for a thorough discussion. See also Battarbee et al. 2005 [https://doi.org/10.1016/j.gloplacha.2004.11.007] for an explanation of the nonlinear nature of the diatom signal in Lake Baikal sediments due to the dissolution of ~99% of diatoms prior to their burial in Lake Baikal sediments.

L126–129: After looking at this paleotemperature record again, the claim made here about Lake El'gygytgyn is a stretch. This is reflected by the weak p-value, which does not compel me to think the "trend" discussed here is statistically significant. This statement also appears to ignore the pollen paleotemperature record from Lake El'gygytgyn, which shows a long-term cooling (see Fig. 3d of Daniels et al., 2021 https://doi.org/10.1002/jqs.3347).

L135: I do not understand what "or (and)" means.

L157–159: I question whether this reference is appropriate. The study here is a discussion of spatial patterns of seasonal temperature change in Eurasia, which is not really relevant to the discussion of whether warm and cold season temperatures can evolve in the same direction that it is invoked to support in this manuscript.

L146–163: This discussion seems to largely miss an important point: we fundamentally do not understand the seasonality of the brGDGT temperature proxy. See discussion in Zhao et al., 2023 [https://doi.org/10.1016/j.quascirev.2023.108124]. In the tropics this is not a problem since seasonality is small, but in the middle and high latitudes this leads to a weak relationship between MAF temperature and MBT'5Me (Zhao et al., 2023).

L168–173: This is hard to follow, please clarify.

L174–177: Again, I urge caution in interpreting the Lake Baikal biogenic silica record as a paleotemperature proxy. See my comment above.

L222–224: Tropical temperatures in China and Africa, not terrestrial temperatures in toto.

Figure 1: Please make the arrows a color other than bright green on the red background. The arrows are not visible to colorblind people.

Figure 2: It is not necessary to overlay your temperature record onto the other records. They can be compared simply by looking at the differences in their trends.

Extended Data Figure 4: Please choose a colorblind figure color palette for panel B.

Extended Data Figure 7: Wouldn't the topical calibration of Zhao et al. (2023) be more appropriate here? The global calibration is intended for deep time applications.

Version 1:

Reviewer comments:

Reviewer #1

(Remarks to the Author)

Dear authors,
after reading the revised version of your manuscript, I can conclude that you have addressed all of my comments and

revised the manuscript accordingly. I have no further concerns or questions.

Yours sincerely,
Jiri Laurin

Reviewer #2

(Remarks to the Author)

I appreciate the authors answering my concerns. However, I don't think the main concerns of how the data is presented was solved and I think the message of the manuscript needs a revision after the reviews that have already been done. Additionally, there is no discussion of how the high elevation (>1000m) of the archive can affect the temperature seen in the record, which is fundamental for interpretations of land temperature reconstructions.

Even though you think the non-thermal error is small because of the lithology of the core you don't have a way to quantitatively address it. I think what you need is an appropriate propagation of error in your study to report your warming temperatures, and then evaluate if it is statistically significant. You NEED to incorporate the RMSE of the calibration you are using.

You mentioned a "long term" warming in your abstract, however, there is no quantitative result of the warming or any temperature reconstruction. I encourage you to add a quantitative result with the appropriate propagation error to make your abstract clearer and easier for readers to assess the impact of the study.

And in response to your reply of my first comment, I am glad that you show the monte carlo simulation of the slope, but it still doesn't show the appropriate error propagation of the brGDGT-based temperature reconstruction. I want to see the appropriate lower and upper boundary of your results.

Lake Baikal: seems to me that after answering my concern this is not a useful record to show in your study. Don't show data that doesn't support your hypothesis or the main point of the paper. It's a distraction. It is not improper to not include data in your figures that doesn't help with the story you are telling. You are more than welcome to mention them and include them in suppl. Material but it distracts in short format papers.

Lake Malawi: 2.9°C/Myr, please add the uncertainties and appropriate error propagation for this. And same comment as Lake Baikal.

Now that you clarify the elevation and included such data... What about changes in lapse rate? How this high-elevation (above 1000m) record is going to amplify the regional temperature trend? Is the warming that you are seeing an amplification of temperature due to elevation and not a response of the mechanism? I think you need to expand more on this and explore the elevation-dependent warming or amplification/changes in temperature. I suggest you do it in a way that you can differentiate if you 1°C is a regional warming or a temperature amplification due to elevation.

Lines 164-171: please show such ratios in a supplementary figure for all samples to assess how different they are. Please add the references for which those ratios are used to exclude GDGTs that are influenced by "non-thermal" factors.

A suggestion for your Fig. S12. use a red star without filling, not a solid point, so readers can look at the color at your site. Interesting that you argue that that map is consistent with your data but seems inconsistent with other terrestrial data that you showed.

Also, your Figure R3. works as a better data visualization for what you want to say... maybe consider including it in the suppl.

I suggest revisiting the figures you are showing in the main text and focus on the data that helps you support your argument (or main message of the paper). Right now for me looks confusing.

Under data availability: what is figshare? Where is the data going to be stored for open access? Make sure you include each individual GDGT (fractional abundance, branched and iso) on it.

Reviewer #4

(Remarks to the Author)

Wang et al. presents a 2-million-year temperature records from Heqing Paleolake in southwestern China. The main findings include a warming trend, instead of cooling, during the time interval of 1.8 – 0.6 Ma. The authors link this to the growth of the Antarctic Ice Sheet. The revision is well-written, and I enjoy reading this story.

I joined the peer review process after the first round and mainly evaluated the revision/supplements and the rebuttal letter, especially the communications between the authors and reviewer #3. The main concern centers on the insufficient assessment of the non-thermal effects on the brGDGTs, thus will undermine the use of brGDGT as a temperature indicator in this study. After carefully reading the revision, I feel the authors have largely incorporated the Reviewer's comments and have added a thorough analysis to properly discuss the potential non-thermal influence. I do believe a rigorous assessment of the brGDGT is a must for such an important and long terrestrial temperature record, and I appreciate the authors' efforts to

address those questions.

I want to echo one important concern that was raised by Reviewer 3 regarding the calibration used in this study. The authors have done a good job of arguing that the brGDGTs are mainly influenced by temperatures, and thus can be a good temperature indicator. What is missing (in the paragraphs around line 125) here is to justify that the best calibration is the one from Wang et al. (2024) EPSL. I encourage the authors to add some discussions here, and I believe it's not hard, given that the Supplementary Figure 13 shows very similar patterns using different calibrations. The authors briefly touch on this in the method section, but I feel some discussion in the main text is essential. So far, the authors have spent quite a lot of time ruling out the non-thermal effects in the first subsection, and this is the time to directly add information that brGDGTs can be used to reconstruct temperature successfully with this calibration. A few sentences stating the advantages of this calibration and why it fits the best will be very helpful.

The figures showing soil GDGTs are missing in the supplements. The authors call out Supplementary Fig. 2 in Lines 84, 88, and 92, but that is not the correct one.

Line 101, regarding the GDGT-0/cren, please consider citing Schneider et al. (2024) QSR. DOI: <https://doi.org/10.1016/j.quascirev.2024.108851>

Version 2:

Reviewer comments:

Reviewer #2

(Remarks to the Author)

I thank the authors for answering all my concerns. I think the paper is now ready for publication with only some minor comments that they will need to add before acceptance.

Please add the 95% confidence interval result in your abstract (see Tierney et al., 2025 AGUAdvances as an example of how to report it).

The first sentence of your abstract should be something that highlights the importance of the period you are reconstructing... not just saying that we don't know terrestrial MAT. Why is it important to reconstruct terrestrial temperature in the Pleistocene? (again, you can look at the previous paper as an example)

Line 35, replace with "Earth's climate shifted from 41..."

Line 233: non-elevated = than those near sea-level

Line 234: it makes no sense to start that sentence with "on the other hand" when you are talking about another period of time, maybe you can say: "During the Last glacial maximum, GDGT-based temperature reconstructions from East African lakes show amplified cooling with elevation..." both modern and LGM show that at higher elevation temperature amplification is bigger than near sea level.

about the data availability: the link that was provided to me does not show the fractional abundances of branched and iso GDGT or neither the ratios calculated in this study. It only shows the Age (Ma), MAT, and MAT-error. PLEASE make sure to upload all of the data so your study is reproducible.

(Remarks on code availability)

yes it has a readme and instruction of how to install and run the code.

Reviewer #4

(Remarks to the Author)

My concerns have been resolved after reading the revision and the response letter.

(Remarks on code availability)

Point-by-point response to the reviewers' comments

We greatly appreciate the 3 reviewers for constructive comments concerning our manuscript entitled “Pleistocene terrestrial warming trend linked to Antarctic Ice Sheets growth” (NCOMMS-24-85101-T). We have addressed those comments carefully and made corrections or explanations accordingly. The following details our point-by-point responses (in blue) to those comments, while in the main text, our changes are noted by “track changes” (line numbers mentioned here refer to the marked manuscript). We are grateful for those comments and suggestions that further improved the quality of our work.

Reviewer #1 (Remarks to the Author):

The manuscript by Wang et al. presents new estimates of terrestrial mean annual temperatures (MAT) from a low-latitude site of China covering the last 2 Ma. The results add to the growing list of arguments for a long-term decoupling of regional terrestrial temperatures from benthic temperature trends and changes in ice volume during the Middle Pleistocene. The novel aspects of this study include namely the geographic location of the dataset, which represents the first MAT estimate for the low latitudes of Eurasia. The low latitude setting makes this dataset valuable due to a relatively weak seasonal bias (aliasing), but this part of the manuscript needs a more careful argumentation, in my opinion (please see “specific comments” below).

The manuscript is well written; a few typos and stylistic issues are listed below. I would also suggest improvements to the structure of argumentation (please see below).

Methods are described adequately, in my opinion, and conclusions appear supported by the data, although I am not able to fully evaluate veracity of the GDGT proxy due to my limited expertise. The hypothesized mechanism of the long-term temperature decoupling builds upon previous literature (which is referenced), and is partly supported by paleoclimate model in the supplement.

The results represent an important contribution to our understanding of the climate evolution and high–low latitude interactions during the Pleistocene. I enjoyed reading about the proposed implications for the migration of hominins to Eurasia.

Re: Thanks for your positive assessment of our work, and please see below our response to your comments.

SPECIFIC COMMENTS:

1) brGDGT proxy at Heqing

The authors propose that the low-latitude lacustrine record of brGDGT is not strongly affected by seasonality (line 72), which seems reasonable given the low latitude setting. However, considering the sensitivity of MAT to winter temperatures at the Heqing site (line 105), I would expect some discussion on the origin of brGDGT in the context of seasonal variations in temperature and precipitation. The discussion on lines 146-163 appears incomplete.

Please explain:

are the GDGT-bearing bacteria autochthonous to the lake or were the lipids transported to lake from soil profiles?

Re: We further collected and analyzed soils surrounding the Heqing Basin. The results indicate that brGDGTs were predominantly produced within the Heqing paleolake, and the influence of soil input is minor.

First, in surrounding soils (Supplementary Fig. 1) brGDGTs are characterized by higher degree of methylation (MBT) and lower degree of cyclization (DC) than those in lake sediments in the Heqing Basin (Supplementary Fig. 2). Therefore, while potential soil brGDGT input can bias MBT and reconstructed temperature towards higher values, DC should be biased towards lower values. However, DC is weakly but positively correlated with MBT ($r = 0.14$, $p = 0.01$) and not correlated with reconstructed MAT ($r = 0.06$, $p = 0.22$) in our record (Supplementary Fig. 3).

Second, the ratio of hexamethylated to pentamethylated brGDGT (IIIa/IIa) can also be used to distinguish brGDGT provenance, and a value > 0.92 can generally indicate aquatic origin of brGDGTs (Xiao et al., 2016, BG), although a value < 0.92 not necessarily means significant soil contribution in lakes. IIIa/IIa is > 0.92 for 97% samples in the HQ core (avg. 1.56 ± 0.38 , $N = 380$), much higher than that in surrounding soils (0.10 ± 0.05 , $N = 12$).

Nevertheless, to rigorously constrain the potential effect of soil input and water-column structure, we further exclude samples with IIIa/IIa < 0.92 or GDGT-0/cren > 1.5 . Only 23 samples (accounting for 6% of the 380 samples) were removed, and therefore our previous record and interpretation was not affected.

Please see Lines 95–110 and Lines 163–170 for related discussion.

What is the role of monsoonal seasonality in the growth and preservation of these bacteria/lipids? I understand the argument with Tsuga content on lines 154-157, but considering that this winter proxy varies on glacial/interglacial time scales, is it reasonable to neglect seasonality as one of the controls on the GDGT dataset? Please try to address these questions in the discussion.

Re: In soils, due to the effect of soil humidity on brGDGT production, seasonal changes in monsoon precipitation might affect the brGDGT palaeothermometry. In the Heqing paleolake, however, as brGDGTs were autochthonous, brGDGTs should not be affected by seasonal changes in water availability. We can further neglect temperature seasonality as a dominant control on the GDGT dataset. Given the low latitude setting, the Heqing region experiences restricted monthly changes in air temperature (Supplementary Fig. 1), and when further considering the buffering effect of lake water, seasonal temperature changes might be even smaller. More importantly, it is generally believed that brGDGT-producing bacteria is active above freezing and therefore brGDGTs can record mean temperature above freezing (MAF) or mean lake water temperature (MLWT) (Zhao et al., 2021, SB; Martínez-Sosa et al., 2021, GCA; Zhao et al., 2023, QSR; Wang et al., 2024, EPSL). At Heqing, the relatively high winter air temperature ($7.5\text{ }^{\circ}\text{C}$) suggests that lakes in this region should be ice-free year-round, and consequently MAF or MLWT equals MAT. Two lines of evidence further indicate that the growth and preservation of these bacteria/lipids might not depend on seasonal temperature variations in non-freezing lakes: (i) in equatorial lakes spanning a MAT range of ca. $2\text{--}25\text{ }^{\circ}\text{C}$, no correlation between brGDGT concentration and temperature is observed (Tierney et al., 2010, GCA), and (ii) in the global lake dataset, MBT'_{5ME}

exhibits better correlations with MAF or MLWT than with mean summer temperature (MST), and additionally, reconstructed growth temperature approximates MAF or MLWT rather than MST (Supplementary Fig. 6). This has been discussed in Lines 145–161 in the revised manuscript.

2) Structure of argumentation

The section “Possible controls...” (lines 146-..) includes the main discussion of the mechanism of temperature decoupling. The first paragraph, addressing the possible seasonal bias in the GDGT proxy, does not fit here. I would suggest moving this paragraph higher, as a separate section (perhaps “The role of seasonality...” or “Seasonal bias...”?).

Re: Thanks for this suggestion. This paragraph is removed and the discussion of seasonality is added in Lines 145–161 in the “Assessing non-thermal influence on the brGDGT paleothermometer” section.

3) Language

L20-24: this is a terrible sentence; please rephrase, split in two

Re: This sentence was deleted due to the word limit in Abstract.

L36: “flourish” is a verb; please rephrase “facilitated the flourish”

Re: Thanks, and “facilitated the flourish of archaic humans” is changed to “been beneficial for archaic humans’ flourishing” (Lines 39–40).

L46: perhaps not necessary to explain that we inhabit the terrestrial realm.

Re: We have deleted “where we inhabit” according to your suggestions.

L90: “full” instead of “fully”

Re: Revised (Line 191).

L132-136: sentence too long, hard to understand; please rephrase

Re: This sentence was split in two (Lines 250–255).

L192: “Progressive” instead of “progressively”

Re: Revised (Line 308).

L199: “albeit quite qualitatively”: what do you mean? please explain

Re: We meant that the model results might be simplistic and cannot exactly capture the real processes in AIS expansion. This is explained in Lines 320–322.

Reviewer #2 (Remarks to the Author):

Wang et al. show a new Pleistocene (2 - 0.2 Ma) terrestrial temperature reconstruction from a paleolake in southwestern China using fossilized microbial lipids (brGDGTs). The study presents a valuable high-resolution temperature record from a region and an archive (land temperatures) where there is little long-term continuous Pleistocene temperature reconstruction. They hypothesized that Antarctic Ice Sheet expansion is the cause/mechanism of a warming during the MPT in terrestrial archives, which they differ

from Global SST. However, I don't think that the hypothesized mechanisms presented here, the references of additional studies to support their hypothesis and the extended data support their study/hypothesis. I have a lot of main concerns and I think a re-evaluation of what I think is an over-interpretation of the data needs to be done.

Re: Thanks for your comments which help us further improve our manuscript. Please see below our responses to your comments.

Main comments:

My main concern is the over-interpretation of the temperature change seen in the record. When taking into account the RMSE of the calibration and standard deviation of the data, it's hard to see how this warming can be separated from the natural variability of the reconstructed temperature. A 1°C temperature change from a proxy that has 1.8°C (keep in mind this is +/- 1.8°C) it's a big stretch in interpreting significant temperature changes to claim such mechanisms/conclusions.

Re: A Mann-Kendall trend test ($S = 4148$, $Z = 3.16$, $p < 0.01$) suggests that the warming trend (1.0 ± 0.3 °C/Myr) is statistically significant. That is, the warming trend is not a random noise, and along with the clear glacial/interglacial cycles, could not be induced by the calibration uncertainty. The 1.8 °C calibration error is caused by the uncertainties in obtaining growth temperature for each sample in the calibration set, as well as the global spread of samples and the accompanying variation in other non-thermal environmental parameters. At a single sit and for a specific core, this systematic error should be much smaller, particularly for the HQ core where non-thermal influence on the brGDGT paleothermometer can be well-constrained. Similar discussion has also been presented by many GDGT studies such as Peterse et al. (2011, EPSL), Tierney et al. (2010, NG), and Johnson et al. (2016, Nature). Therefore, for our temperature record, while absolute temperature estimates should be interpreted with caution, we believe that relative temperature changes are generally reliable. Actually, other calibrations based on various statistic methods yield similar warming trend from 1.8 to 0.6 Ma, ranging from 0.7 to 1.6 °C/Myr (avg. 1.1 °C/Myr), further suggesting that the warming observed here is robust. The related discussion has been added in Lines 197–199 and Lines 445–461. Moreover, we performed Monte Carlo simulations to assess uncertainty on the temperature record for 1.8–0.6 Ma. We added an error of 1.8 °C to see whether this uncertainty would affect our conclusion. The results for 1,000 times of simulation showed that 99.9% of the slopes were negative (as the slopes were calculated from younger to older ages, Fig. R1), confirming that the observed warming trend during 1.8–0.6 Ma is reliable.

Figure R1. Distributions of slopes under 1,000 times of Monte Carlo simulation (with an uncertainty of 1.8 °C) for the Heqing temperature record during 1.8–0.6 Ma.

Second, I am concerned about the references. First, they are not only wrong (the number doesn't correspond to the reference), but also I could not find the same data or information reported by the references in some cases.

Re: Many thanks for pointing out this and we have carefully checked the references. We are sorry for the incorrect reference numbers in the figure captions and have corrected them in the revised version. The reference numbers in the main text and the data used in the previous version are correct.

1. Lake Baikal reference number is wrong, and the closest reference of it doesn't show the BioSi reconstructions that are shown here in Figure 3. The paper uses Biogenic Si to identify glacial and interglacial sections but it has not been used as a long-term temperature reconstruction. This is mostly because BioSi is subjected to dissolution, so you can't use the abundance of it to say that there is a temperature trend over time, you use it to identify warmer intervals than others to then do cyclo-stratigraphy or studies like that, not to infer quantitative temperature changes over time.

Re: Here we refer to Prokopenko et al. (2006, QSR), and the data were downloaded from <https://www.ncei.noaa.gov/access/paleo-search/study/6068>. The dissolution of BioSi mainly occurs above the upper 30cm of sediments, in the water column or at the surface sediment-water interface. In deeper sediments, Si saturation is reached, thereby buffering against further dissolution (Mackay, 2007, ESR). This implies that dissolution should not significantly affect long-term trends of BioSi. Actually, for the 5-Ma diatom record of Lake Baikal (Prokopenko et al., 2001, Q1), it generally resembles the benthic $\delta^{18}\text{O}$ record before 1.8 Ma rather than exhibiting an increasing trend (Fig. R2). This further indicates that the long-term trend of diatom and BioSi should not be significantly biased and the relatively stable trend during the past 1.8 Myr might record a climatic signal. Moreover, we acknowledge that the dissolution effect is likely to have had a relatively large impact on the preservation of diatom valves during glacial periods because of overall low diatom productivity (Mackay, 2007, ESR). However, the interglacial trend might be generally reliable. Nevertheless, considering your concerns, we have revised this section as “The biogenic silica (BioSi) record from Lake Baikal (54 °N), which probably tracks temperature fluctuations in high-latitude continental Eurasia, shows no obvious trend from 1.8 Ma to 0.6 Ma (Fig. 3d). However, it is generally believed that the response of BioSi to temperature is non-linear and qualitative, especially due to the significant dissolution effect during glacial period. Therefore, only the interglacial trend might be reliable” in Lines 231–236.

[REDACTED]

Figure R2. [REDACTED]. Prokopenko et al. (2001, QI).

2. Lake Malawi, the proxy used here is TEX₈₆ which has its own issues in freshwater lakes, but more importantly, the original paper doesn't claim to have an increase in temperature because the variability in the data is within the uncertainty (2sigma = 0.8°C). Additionally, this study clearly showed CO₂ as the main driver of significant temperature changes along the record and the overall conclusion of it had to do more with a wetter/drier climate than temperature changes.

Re: The TEX₈₆ paleothermometer is generally problematic in lakes, but it might be reliable in large lakes such as Lake Tanganyika (Tierney et al., 2010, NG) and Lake Malawi (Johnson et al., 2016, Nature), especially with rigorous quality control. For this record, samples that may potentially be affected by non-Thaumarchaeota contribution and soil input have been rejected for analysis. The remaining accepted temperatures indicate a 2 °C warming from 1.3 to 0.6 Ma, larger than the 2 sigma error (0.8 °C). We agree with the reviewer that this study mainly focus on long-term hydrological rather than temperature changes, and for temperature only glacial-interglacial changes for the past 0.6 Ma were discussed. However, given that no other long Pleistocene terrestrial temperature record is available from Africa, we feel that it is not proper to ignore this work. Nevertheless, to address your concern, we have revised the discussion on the Lake Malawi record as “Beyond Eurasia, a 1.3-Myr-long temperature record inferred from TEX₈₆ at Lake Malawi, East Africa (11 °S) shows an approximate 2.9 °C/Myr warming trend ($p < 0.01$) during 1.3–0.6 Ma (Fig. 3e). However, for this record, the

TEX₈₆ paleothermometer might be complicated by the ill-defined ecology of freshwater *Thaumarchaeota*, although some samples that were potentially affected by non-Thaumarchaeota contribution and soil input have been rejected for analysis” in Lines 243–249.

3. Lake El'gygytgyn: this warming rate you mention here 1. Is not significant (see p-value) and the record doesn't show cooling around the MPT (in agreement with your hypothesis) but also it does not show warming. The main change that happens at this high-latitude site is a change in hydrological conditions.

Re: We fully agree with you, and this sentence is revised as “Further north at Lake El'gygytgyn, Far East Russia (67.5 °N), while regional aridification increased during the MPT, no significant warming or cooling trend is observed based on brGDGTs” in Lines 236–239.

4. In line 435, reference 24, this paper doesn't show anything about the formation and demise of the Heqing paleolake. I could not find such a reference. Also, I don't know why you dismissed those intervals? Even though they don't represent a full lake according to what you said they could give you more information about changes during key periods of time (last interglacial to LGM and Plio-Pleistocene) that have been shown to have a more extreme or a more drastic change that could help differentiate noise vs. signal.

Re: Sorry for the incorrect reference number cited here. We intended to cite An et al. (2011, Science) which stated that “an abrupt drop in $\delta^{18}\text{O}$ (a measure of the ratio of stable isotopes $^{18}\text{O}:^{16}\text{O}$) and a shift in the relationship between $\delta^{18}\text{O}$ and $\delta^{13}\text{C}$ (a measure of the ratio of stable isotopes $^{13}\text{C}:^{12}\text{C}$) of adult ostracod *Ilyocypris microspinata* shells around 11.1 m probably indicate the beginning of a change from a closed paleolake to an open system. At 7.4 m depth, the simultaneous disappearance of two benthic ostracod species (*Ilyocypris microspinata* and *Lineocypris jiangsuensis*) indicates that the basin became a fully open system as it is today. Accordingly, we focus on the interval when the system was closed”.

In the revised manuscript, we have added GDGT data from the upper section, and it now covers the 0–2 Ma. However, 23 samples were rejected after carefully assessing the potential effects of non-thermal factors such as soil input and changes in water-column structure on brGDGTs. The excluded samples are mainly from the top section of the core, in agreement with our previous statement. This was discussed in Lines 164–171.

Despite that brGDGTs cannot be used to reconstruct temperature during the past 0.15 Ma at the Heqing paleolake, a previous study from another site in southwestern China has demonstrated that brGDGTs can quantitatively track past temperature variations from last interglacial to LGM in this region (Zhao et al., 2021, SB). Moreover, we wish to focus on the interval after Northern Hemisphere Glaciation (2–3 Ma), especially the long-term trend from 1.8 to 0.6 Ma, and therefore samples older than 2 Ma were not analyzed. Nevertheless, the close match of our biomarker-inferred MAT with marine SST records on orbital timescales, and with Tsuga content in both glacial-interglacial variation and long-term trend, reinforce the robustness of our quantitative temperature estimates.

5. Reference 81 and 82. These are not peer-review articles. 81 is a public-facing article from a science journalist and 82 is a conference abstract. I don't think you should cite

non-peer review references in a manuscript submitted to Nature Communications.

Re: Thanks for this reminder. We have deleted the two references in the revised manuscript. They were used to argue for our CO₂ stack and therefore this does not affect our temperature record and related discussion.

Overall, we acknowledge that the records from Lake Baikal, Lake Malawi and Lake El'gygytgyn might have their own weakness and limitation. However, it appears to be improper to ignore all these records due to the lack of Pleistocene terrestrial temperature records with higher quality in other regions. In the revised manuscript, we have just described these records and pointed out potential complicating factors. We also avoid concluding a long-term pattern for other regions beyond East Asia due to limited evidence. Actually, the large uncertainty in other terrestrial records highlights the importance of our Heqing MAT record.

Third, one of your data that supports your hypothesis is your extended data figure 6 which is the results of the model output of PI-AIS(PI). First, I think the figure should benefit from having labels (“a”, “b”, etc...) and a more descriptive figure caption. Second, the results in this figure do not clearly support your hypothesis or the figures should be presented differently. I think you could benefit from having a box or an arrow that can show what you mean in the text. For example, you have “shows that the expansion of AISs can potentially increase tropical zonal SST gradient”, where in the figure you can see that? I don't see these maps and it is obvious to me to see such an increase in tropical zonal SST. Also, you have “elevate surface temperature over Eurasia” which regions are you talking about because there is a combination of warm and cold regions over Eurasia. You have “with an overall increase in water vapor in the Asian monsoon region”, again, highlight the regions you are talking about in these maps because I see some strong red colors over India/Tibet but a blue color around South China where I think your core comes from. Anaways, I don't see how these results clearly support your statements. By the way, next to the water vapor panel there is an “x” above 30N, what's that?

Re: Thanks for these comments. According to your comments, we have further revised the figure for modelling results (Supplementary Figure 12) and enrich its caption to support our statements.

(i) The labels for each panel and a more descriptive figure caption were added in this figure.

(ii) We used a blue box and a green arrow to highlight a decrease in SST in the eastern tropical Pacific relative to the western tropical Pacific, and thus the strengthening of the zonal SST gradient in the tropical Pacific in panel b.

(iii) In panel a, the terrestrial temperature difference between PI and AIS experiments shows an overall warming in the majority of the Eurasian continent (except Eastern Russia and India) driven by AIS expansion. The warming at Heqing (red circle) is consistent with proxy-based reconstruction.

(iv) The northward penetration of summer precipitation and water vapor in central and northeastern China (panel c and d) suggests intensified Asian summer monsoon driven by AIS expansion, despite that summer precipitation and water vapor might be reduced in southeast Asia. This might be more obvious if only the Asian land monsoon region was analyzed (Fig. R3).

(v) The “x” next to panel d was deleted.

Overall, despite that our model might be simplistic and may not exactly capture changes in climatic parameters, it demonstrates that the expansion of AISs can potentially increase tropical zonal SST gradient and elevate surface temperature over the majority of the Eurasian continent, with an overall strengthening in Asian summer monsoon.

Figure R3. Differences in summer precipitation rate and summer water vapor between PI and AIS experiments over the Asian land monsoon region. The range of the monsoon region is defined by Wang et al. (2011, CD).

Minor comments:

Add elevation in abstract? I think that when talking about temperature reconstruction it's good to know if the terrestrial temperatures are near sea level or high elevation sites.

Re: Its altitude is 2190 m a.s.l., and we have added it in the Abstract (Line 27).

Add modern MAT in the figures to see how it compares with your reconstructed temperatures? The general thought is that the last million years were colder than today... So I want to see how modern MAT looks with your reconstructed temperature and if your estimate is colder or warmer than that as your present absolute values.

Re: The modern MAT is 13.8 °C at the Heqing Basin (Line 378). As is expected, it is generally higher than reconstructed temperatures for the last million years. However, as explained previously, absolute temperature estimates should be interpreted with caution, and relative temperature changes are more meaningful. Therefore, we have not discussed this in the manuscript.

You used the Global SST from Clark, but they also present SST compilations by basin. I think that rather than using individual cores near your location you can add how the SST evolution in the Warm Pool looks in comparison with your record. Then, you can evaluate other mechanisms of why you don't see a clear temperature change over the MPT like other global records.

Re: Thanks for this suggestion. We have added the SST stack for Indo-Pacific warm pool (IPWP) in Supplementary Fig. 10 according to your suggestion. The IPWP stack also shows a relatively stable temperature during 1.8–0.6 Ma, in contrast to the significant cooling in other basins such as eastern tropical Pacific. This is in agreement

with our hypothesis (discussed in Lines 302–305) that the expansion of AISs can potentially increase tropical zonal SST gradient and promote relatively increased heat accumulation in the western Pacific, then strength Asian monsoons, and finally elevate surface temperature over Eurasia.

Line 186-187: you are claiming that there is a decrease in SST in the equatorial eastern Pacific that is strengthening the tropical zonal SST gradient and walker circulation, but when looking at Figure 4c.. 1. I think the records should go all the way back to the beginning of the Pleistocene, and 2. you can have that data with the new Global SST compilation by basin from Clark et al. 2024.. And probably highlight where the “pronounced decrease” happens because with my eyes, in that figure, I don’t see any “pronounced” change.

Re: Many thanks for this recommendation. We have recalculated the zonal SST gradient using SST stacks for Indo-Pacific Warm Pool (IPWP) and Eastern tropical Pacific (ETP) reported by Clark et al. (2024, Science). This extends the zonal SST gradient record back to 2 Ma. We have also added the ETP SST stack of Clark. Please see Fig. 4c for the two records. Data older than 2 Ma are not presented as they are beyond the scope of this work.

You report a 1°C/myr... but what's the error or standard deviation of this?

Re: It is 0.3 °C/Myr and we have added it in Line 198.

Line-by-line

Line 60-61: “This implies that Pleistocene global temperature, or at least terrestrial temperature, cannot be simply represented by marine temperature.” I think you should re-arrange this sentence by saying something like global temperatures cannot be represented only by marine archives and the incorporation of terrestrial temperatures is needed for assessing global temperature changes and trends. Or something along the lines that global temperature are both terrestrial and marine, not that terrestrial temperatures are not represented by marine (that is obvious)

Re: Thanks, and this sentence is changed to “This implies that global temperatures cannot be represented only by marine archives and the incorporation of terrestrial temperatures is needed for assessing global temperature changes and trends” in Lines 65–67.

Line 62: Do you mean glacial-interglacial? Or just glacial? Is just strange to only mention glacial if you are referring to the entire pleistocene. Also that sentence could be two sentences rather than one.

Re: We intended to highlight glacial periods here, as brGDGTs might be strongly affected by vegetation coverage and seasonality in soils at the mid-latitude CLP. This sentence is revised as “However, brGDGTs are argued to be likely affected by vegetation coverage and seasonality in soils at the mid-latitude CLP, especially during glacial periods” (Lines 69–71).

Line 69: what's the elevation?

Re: The elevation (2190 m a.s.l.) is added in Line 78.

Line 80: looking at the SD Fig.1. Seems to me that the gradual transition really starts at ~1.2Ma in your record... not before... no?

Re: According to your suggestion, this sentence was changed to “On orbital timescales, the 100-kyr glacial-interglacial cycles gradually replaced the 41-kyr cycles, with strong periods of both 41 kyr and ~80–120 kyr during the Middle Pleistocene Transition (MPT, 1.25 to 0.7 Ma) (Supplementary Fig. 7)” (Lines 179–182).

Line 89-92: is there a reference to cite here? As it is written it seems like there is a need for a larger evidence of that high-latitude ice-volume forcing.

Re: Here we have added “Clark et al., 1999, Science”, and changed “high-latitude ice-volume forcing” to “ice sheet influence” (Lines 192–193).

Line 197: reference?

Re: A reference (Allen and Ingram, 2002, Nature) is added (Line 314).

Line 704: what is ESL, I don't see it defined anywhere.

Re: Its definition “ice-volume equivalent sea-level” is added in Line 1077.

Reviewer #3 (Remarks to the Author):

Review of Wang et al. for Nature Communications

Joseph B. Novak

Recommendation

Reject and encourage resubmission.

Summary

Wang et al.'s manuscript entitled “Pleistocene terrestrial warming trend linked to Antarctic Ice Sheets Growth” presents a 2-million-year terrestrial paleotemperature record from Heqing paleolake in southwestern China. The reconstructed paleotemperatures show a warming trend, rather than cooling, over the 1.8–0.6 Ma interval, in contrast to marine sea surface temperature records, which the authors attribute to dynamical changes in the Earth's atmospheric circulation in response to Antarctic Ice Sheet growth.

The lack of a rigorous assessment of the branched GDGT biomarker distributions used to generate the paleotemperature record is seriously concerning. While the authors argue that the similarity between the reconstructed paleotemperatures to a pollen record in the same sediment core indicates that the brGDGTs are primarily recording temperature, I remain unconvinced, especially since this core comes from what appears to have been an alkaline lake based on the occurrence of carbonate sediments (Shen et al., 2007; ref. 52 in this manuscript). The presentation of the pollen data on a log-scale is somewhat odd and, possibly, misleading since pollen data are usually presented on a linear y-axis scale.

A thorough assessment of any possible non-thermal influence on the brGDGT distributions in this lake is fundamental to this study. The lack of this needed supplementary discussion ultimately motivates my recommendation to reject the manuscript. However, the data presented here hold the potential to be a landmark dataset in the field if the authors can show that these non-thermal factors are not influencing the brGDGT biomarkers here. Therefore, I encourage the resubmission of

this manuscript after my comments are addressed.

Re: Thanks for your positive assessment of our record. We agree with you that a thorough assessment of any possible non-thermal influence on the brGDGT paleothermometer in this lake is fundamental to this study. Accordingly, we have added a new section to discuss the potential influences of soil input, changes in water-column structure, water chemistry, and bacterial communities, and seasonal bias in details (Lines 91–171). We can demonstrate that these factors are unlikely to significantly bias our temperature record, especially when excluding 6% samples (mainly from the top section of the core) that may potentially be affected by soil input and water-column structure changes. We hope that the supplementary discussion can address your concern about the robustness of our temperature record.

We should point out that the high carbonate content in the core is because that the Heqing basin is a mountainous basin with regional bedrock being dominated by carbonates (An et al., 2011, Science; Yang et al., 2024, QSR), rather than indicating a brackish to saline paleolake. Actually, the occurrence of freshwater diatoms (Yang et al., 2024, QSR), and negligible late-eluting isomers after 5-methyl and 6-methyl brGDGTs which are generally high in brackish and saline lakes (Wang et al., 2021, GCA) throughout the core suggest a freshwater environment. In the global lacustrine brGDGTs dataset for freshwater lakes, while most lakes are alkaline, brGDGTs can faithfully track temperature and there is no significant offset in reconstructed temperature between lakes with pH = 7–8 and > 8 (Supplementary Fig. 5). Therefore, we believe that the high carbonate content (and possibly high alkalinity) do not affect the quantitative application of the brGDGT paleothermometer in Lake Heqing sediments. This supplementary discussion is added in Lines 122–130. Moreover, according to your suggestion, the pollen data are presented with a linear y-axis scale (Supplementary Fig. 9b). Nevertheless, the overall pattern remains unchanged, and its long-term trend and glacial-interglacial variation both align with those in our reconstructed MAT.

Major Comments

Extended Data Figure 3: Why is the Tsuga data plotted on a log scale? This is odd to me since pollen data are typically plotted on a linear scale. Is there a justification for this or is it simply to present that data in such a way that they appear to closely resemble the brGDGT MAT reconstruction? Similarly, is the strength of the correlation shown in the scatterplot dependent on the log transformation of the Tsuga pollen data? Is there perhaps some sort of bulk geophysical proxy data (magnetic susceptibility, bulk density) that could be used to make this comparison more effectively?

Re: A log scale used previously was to reveal the subtle changes in Tsuga during glacial periods (quite low), following An et al. (2011). Using a log scale will not alter the overall structure of the data. In the revised version, Tsuga pollen is plotted on a linear scale in Supplementary Fig. 9 according to your suggestion. The glacial-interglacial variations and long-term trends of Tsuga pollen agree even better with the brGDGT-based MAT reconstruction, and the correlation coefficient (r) in the scatterplot slightly increases from 0.45 to 0.47 (Supplementary Fig. 9c). The magnetic susceptibility record is added in Supplementary Fig. 2j. It provides further support for the substantial changes in lake depositional environment at ~0.15 Ma inferred from other proxies.

L146 and Figure 3: The authors make a statement here regarding “the absence of a long-term terrestrial cooling during the past 2 Myr.” However, the data presented in the

manuscript only represent a relatively small number of locations and narrow band of latitudes. In Figure 3, the authors show data from five sites. One dataset (Lake Baikal) is not a paleotemperature dataset (see my comment below). The remaining four datasets are from the tropics or near tropics, three of which are from China spanning a very narrow range of latitude (27–35°N). So, essentially, are the authors claiming that Southern China and Eastern Africa are representative of terrestrial temperature change throughout the entire Earth? This is unlikely, considering that modern climate change is resulting in remarkably different rates of temperature change in the high latitudes vs. the tropics. See also the pollen paleotemperature record from Lake El'gygytgyn, which shows a long-term cooling since the Pliocene (see Fig. 3d of Daniels et al., 2021 shows this nicely [<https://doi.org/10.1002/jqs.3347>]). Although these data do not sample every glacial-interglacial cycle, they appear to directly contradict the claim made here and are not discussed in the manuscript.

Re: Thanks for pointing out this issue. We acknowledge that the data from 5–6 sites are not sufficient to conclude a global pattern. In the revised manuscript, we avoid the statement that the absence of a long-term cooling during the past 2 Myr is a global phenomenon, and limit our statement to East Asia.

For the Lake Baikal record, it might be improper to ignore this work due to the lack of long Pleistocene terrestrial temperature record in high-latitude regions. In the revised manuscript (Lines 231–236), we have just described the that biogenic silica (BioSi) record shows no obvious trend from 1.8 Ma to 0.6 Ma, and clarified that the response of BioSi to temperature is non-linear and qualitative, especially due to the significant dissolution effect during glacial period, according to your comments. Nevertheless, the overall trend for interglacial might be reliable.

For Lake El'gygytgyn, we find that the pollen paleotemperature record (Melles et al., 2012, Science; Brigham-Grette et al., 2013, Science) not necessarily contradict our claim of the absence of a long-term cooling during the past 2 Myr. Please see Fig. R4 for Lake El'gygytgyn's pollen record compiled by Daniels et al. (2022, JQS). While they reported a continuous record from 3.6 to 2.1 Ma, only data for 4 glacial-interglacial cycles was reported, and the data is especially lacking for the critical period discussed in our study for unknown reason. Therefore, we stated that “High-resolution, but discontinuous, pollen-based Pliocene-Pleistocene temperature record has also been reported for this lake, and data from the critical 1.8–0.6 Ma period is scarce” in Lines 241–243.

[REDACTED]

Figure R4. [REDACTED]. Figure source: Daniels et al. (2022, JQS).

L457–463: Why is there no supplementary text or figures showing the long-term trends in IR_{6Me}, CBT', brGDGT concentration and GDGT-0/Cren ratio in these samples? It is very important to understand the potential for non-thermal factors to influence brGDGT paleotemperature reconstructions, especially in what the authors consider a “terrestrial MAT time series benchmark for Pleistocene terrestrial temperature variations.”

Re: Thanks for this suggestion. GDGT indices such as IR_{6ME}, CBT', and GDGT-0/cren ratio, and brGDGT concentration have been added in Supplementary Data. We have also plotted these GDGT indices in Supplementary Fig. 2 and used them to constrain non-thermal influences (Lines 91–171).

There are several important factors that must be discussed for the community to understand whether the brGDGT distributions are accurately capturing temperature. These include: (1) brGDGT provenance; (2) the potential for changes in lake water chemistry to affect the brGDGT distributions; (3) whether there were changes in other aspects of the brGDGT distributions that are correlated with the warming inferred between ~1.2–0.2 Ma in Heqing Basin.

For example, the brGDGT paleotemperature reconstruction method used here

(developed by the same lead author) is supposed to disentangle the relative contributions of 5-methyl and 6-methyl brGDGT producers when estimating paleotemperatures (i.e., remove the potential temperature bias from multiple brGDGT-producing bacterial communities contributing lipids to sediment). Are the resulting paleotemperature estimates from this method correlated with IR6Me, CBT, BIT, etc. in these samples? If so, does this imply that there is a substantial non-thermal influence on the temperatures reported here that is not accounted for by this method? If not, does that mean the method is working? We do not know because this is not explored.

Re: As mentioned above, according to your suggestions, we have added a new section to discuss the potential influences of soil input, changes in water-column structure, water chemistry, and bacterial communities, and seasonal bias in details (Lines 91–171). We can demonstrate that these factors are unlikely to significantly bias our temperature record, especially when excluding 6% samples (mainly from the top section of the core) that may potentially be affected by soil input and water-column structure changes. We hope that the supplementary discussion can address your concern about the robustness of our temperature record.

Moreover, the positive relationship ($r = 0.33$, $p < 0.01$) between the ratio of 6- over 5-methyl brGDGTs (IR_{6ME}) and MBT'_{5ME} in the HQ core possibly points to a slight impact of bacterial community change related to complex changes in multiple non-thermal parameters (Novak et al., 2025, G3). On the other hand, the reconstructed MAT based on our new calibration has no correlation with IR_{6ME} ($r = 0.08$, $p = 0.14$) (Supplementary Fig. 3). Therefore, we have applied the recent calibration of Wang et al. (2024, EPSL) which may mitigate the isomer effect driven by community shift. This has been discussed in Lines 130–145.

The overarching point of this comment is that there is not sufficient information provided here for me (or anyone else) to judge whether the brGDGT distributions are authentically capturing temperature in the Heqing paleolake record. This is particularly important to consider since the sediments in this lake core are calcareous clays, suggesting that the lake was alkaline. These alkaline lakes appear to be especially challenging environments for brGDGT paleotemperature reconstruction, for example Lin et al. (2024) [<https://doi.org/10.1016/j.earscirev.2024.104694>]. Indeed, Shen et al. (ref. 52 here) report large changes in sedimentary carbonate content in this lake core. Shen et al. described substantial changes in the amplitude of these sedimentary carbonate cycles through time and the deposition of gravels associated with regional mountain uplift throughout the core. Are these sedimentary features, indicative of changes in lake water chemistry and, perhaps, changes in brGDGT provenance, respectively, related to any aspects of the paleotemperature record presented here?

Re: As explained above, in the revised manuscript we have added a new section to discuss the potential influences of non-thermal factors on quantitative temperature reconstruction in the core (Lines 91–171). By carefully screening GDGT distributions, we can demonstrate that these factors are unlikely to significantly bias our temperature record, especially when excluding 6% samples (mainly from the top section of the core) that may potentially be affected by soil input and water-column structure changes. Moreover, the high carbonate content in the core is because that the Heqing basin is a mountainous basin with regional bedrock being dominated by carbonates (An et al., 2011, Science; Yang et al., 2024, QSR), rather than indicating a brackish to saline paleolake. This is supported by the occurrence of freshwater diatoms (Yang et al., 2024, QSR), and negligible late-eluting isomers of 5-methyl and 6-methyl brGDGTs which

are generally high in brackish and saline lakes (Wang et al., 2021, GCA) throughout the core. On the other hand, in the global lacustrine brGDGTs dataset for freshwater lakes, while most lakes are alkaline, brGDGTs can faithfully track temperature and there is no significant offset in reconstructed temperature between lakes with pH = 7–8 and > 8 (Supplementary Fig. 5). As for Lin et al. (2024, ESR), they provide no evidence that alkalinity can affect brGDGT-based temperature reconstructions in lacustrine conditions. Additionally, the calibration method used in our work can effectively mitigate the potential effect of changing isomer ratio related to changes in lake water chemistry and bacteria community structure. Therefore, we believe that the high carbonate content (and possibly high alkalinity) do not affect the quantitative application of the brGDGT paleothermometer in Lake Heqing sediments. This has been discussed in Lines 121–145. Furthermore, the two short intervals of sand layers with fine gravels occurred at 372.5–371.9 m and 195.4–189.5 m (An et al., 2011, Science). For our dataset, only 1 sample was within them, and it exhibits much higher GDGT/cren ratio (2.4) and therefore is rejected for quantitative temperature reconstruction (Supplementary Data).

In my view, this manuscript cannot pass review until the potential non-thermal influence on the brGDGT distributions is addressed. I appreciate the huge amount of effort that went into analyzing this dataset and I strongly encourage the authors to address these comments and resubmit their work. Long terrestrial paleoclimate records are a rare and special thing with the potential to capture the thought and attention of the scientific community. This is why we must treat them carefully, and critically, to be sure about their underlying quality.

Re: We appreciate your positive assessment on the merit of our record. We hope that the supplementary data and discussion can address your concern about its robustness.

Supplementary data: Why is there no supplementary data table with the reconstructed temperatures and underlying brGDGT distributions? It is standard practice to share this information.

Re: Thanks for this reminder and we have added the data in Supplementary Data file.

Minor Comments

L19: “Our knowledge of Earth’s temperature history” rather than “knowledge on.”

Re: This sentence was deleted due to the word limit for Abstract.

L36: “flourishing” rather than “flourish.”

Re: Thanks, and “facilitated the flourish of archaic humans” is changed to “been beneficial for archaic humans’ flourishing” (Lines 39–40).

L42 and elsewhere: The SI convention now is to italicize the δ in $\delta^{18}\text{O}$.

Re: The δ in $\delta^{18}\text{O}$ is italicized in Lines 46, 50, 185, 563, 1051.

L73: timeseries should be written as one word, not two.

Re: “time series” is changed to “timeseries” (Line 86).

L90: “full establishment” rather than “fully establishment.”

Re: “fully” is changed to “full” (line 191).

L93: Consider rephrasing to: “A notable feature of the long-term Heqing paleotemperature trend is that...”

Re: This sentence is rephrased according to your suggestion (Line 194–197).

L96: “trend” in “pronounced cooling trend” can be removed as it is unnecessary.

Re: Thanks, and revised (Line 199).

L104–106: I do not really follow what you mean here. Mean annual temperature by definition incorporates the temperature of all months equally. Is this statement trying to parse the slight difference in the correlation strength between MAT and summer vs. winter temperature? This seems like a bit of a stretch to me since the difference in Pearson’s r here is rather minor.

Re: Here we meant that changes in Tsuga pollen can generally trace MAT variations at Heqing, since (i) Tsuga pollen is largely controlled by winter temperature in this region, and (ii) regional MAT strongly depends on winter temperature. We have revised the two sentences as “Modern investigations show that Tsuga distribution in the Asian monsoon region is controlled by winter temperature at regions with sufficient precipitation. Furthermore, MAT at the Heqing Basin strongly depends on winter temperature (Supplementary Fig. 8). Therefore, Tsuga pollen can be used as a sensitive, albeit qualitative, temperature proxy in this region” in Lines 207–213.

L116–117: does this imply that a different result is obtained if a more restricted pollen training dataset is used?

Re: The overall trend for the pollen-based reconstruction at Zoige Basin might differ when using various modern training-sets with different distances (radius = 500 km, 1000 km, 1500 km, and all samples). However, all reconstructions based on the weighted-average partial least squares (WAPLS) method show a warming trend from 1.5 to 0.6 Ma (Zhao et al., 2021, GPC), when global SST significantly dropped. We believe that the transfer functions using modern training dataset from a broader region might be less affected by non-thermal factors, and therefore cite the record using all modern pollen data from China and Mongolia as the training dataset.

L124–126: The Lake Baikal biogenic silica record does not track paleotemperature directly and certainly not in a linear fashion. I therefore urge extreme caution in interpreting this record as a paleotemperature proxy dataset as is done here. A major drawback of this data type is that it cannot distinguish relative differences in how cold individual glacial periods were. This is because it is impossible to have less than 0% biogenic silica content, meaning that any potential differences in temperature between glacial periods is missed by the proxy signal. You will notice that this is essentially the value of every glacial period (at least the stadials, interstadials have slightly higher content) over the entire 1.8-million-year period covered by the Prokopenko et al. 2006 record you reference here.

See Prokopenko et al. 2001 [doi:10.1006/qres.2000.2212] for a thorough discussion. See also Battarbee et al. 2005 [https://doi.org/10.1016/j.gloplacha.2004.11.007] for an explanation of the nonlinear nature of the diatom signal in Lake Baikal sediments due to the dissolution of ~99% of diatoms prior to their burial in Lake Baikal sediments.

Re: Thanks for this information. For the Lake Baikal record, we fully agree with you that the biogenic silica (BioSi) record does not track paleotemperature in a linear

fashion. However, it might be improper to ignore this work due to the lack of long Pleistocene terrestrial temperature record in high-latitude regions. Therefore, in the revised manuscript (Lines 231–236), we have just mentioned that the BioSi record shows no obvious trend from 1.8 Ma to 0.6 Ma, and clarified that the response of BioSi to temperature is non-linear and qualitative, especially due to the significant dissolution effect during glacial period. Nevertheless, the overall trend for interglacial might be reliable. Further, in the revised manuscript, we avoid concluding a long-term pattern for other regions beyond East Asia due to limited evidence.

L126–129: After looking at this paleotemperature record again, the claim made here about Lake El'gygytgyn is a stretch. This is reflected by the weak p-value, which does not compel me to think the “trend” discussed here is statistically significant. This statement also appears to ignore the pollen paleotemperature record from Lake El'gygytgyn, which shows a long-term cooling (see Fig. 3d of Daniels et al., 2021 <https://doi.org/10.1002/jqs.3347>).

Re: According to your comments and Reviewer 2's, we have revised this sentence as “Further north at Lake El'gygytgyn, Far East Russia (67.5 °N), while regional aridification increased during the MPT, no significant warming or cooling trend is observed based on brGDGTs” (Lines 236–239). For the pollen record, while a continuous record from 3.6 to 2.1 Ma was reported, the data is lacking for the critical period discussed in our study for unknown reason. Therefore, we cannot say that the pollen record exhibits a cooling trend during 1.8–0.6 Ma. We state that “High-resolution, but discontinuous, pollen-based Pliocene-Pleistocene temperature record has also been reported for this lake, and data from the critical 1.8–0.6 Ma period is scarce” in Lines 240–242.

L135: I do not understand what “or (and)” means.

Re: “or (and)” is changed to “or” (Line 253).

L157–159: I question whether this reference is appropriate. The study here is a discussion of spatial patterns of seasonal temperature change in Eurasia, which is not really relevant to the discussion of whether warm and cold season temperatures can evolve in the same direction that it is invoked to support in this manuscript.

Re: The discussion on seasonal bias is moved to an earlier place and the related discussion is deleted.

L146–163: This discussion seems to largely miss an important point: we fundamentally do not understand the seasonality of the brGDGT temperature proxy. See discussion in Zhao et al., 2023 [<https://doi.org/10.1016/j.quascirev.2023.108124>]. In the tropics this is not a problem since seasonality is small, but in the middle and high latitudes this leads to a weak relationship between MAF temperature and MBT'5Me (Zhao et al., 2023).

Re: This section is deleted and we have added a new paragraph to discuss the seasonality of the brGDGT temperature proxy, just limited to the low-latitude Heqing region (Lines 146–162).

L168–173: This is hard to follow, please clarify.

Re: This sentence is revised as “One might argue that this synthesis might mostly rely on records with higher density of datapoints, as its variation resembles that for the

record inferred from $\delta^{13}\text{C}$ of terrestrial C3 plant remains, which accounts for 30% of the total datapoints during 2.6–0.2 Ma (Supplementary Fig. 11a). To reconcile the potential artifact caused by this issue as well as the systematic bias and large scattering of each work using different methods and archives, we further compiled a new CO₂ stack by normalizing 11 long Pleistocene CO₂ reconstructions vetted by the CenCO2PIP Consortium⁵⁰ (Supplementary Methods). The new stack still indicates no long-term trend in CO₂ level during the past 2 Myr (Fig. 4a)” in Lines 275–287. As our two new stacks show rather similar results, we only use the stack based on vetted records in the revised manuscript.

L174–177: Again, I urge caution in interpreting the Lake Baikal biogenic silica record as a paleotemperature proxy. See my comment above.

Re: We have deleted the discussion of Lake Baikal biogenic silica record here.

L222–224: Tropical temperatures in China and Africa, not terrestrial temperatures in toto.

Re: This sentence is revised as “Moreover, our results highlight a possible divergent evolution of land surface temperatures (at least at some regions) and marine temperatures, and the processes involved could be helpful for projecting future temperature changes with regional patterns identified” (Lines 347–350).

Figure 1: Please make the arrows a color other than bright green on the red background. The arrows are not visible to colorblind people.

Re: Thanks for this reminder, and the color of bright green arrows is changed to bright blue (Fig. 1).

Figure 2: It is not necessary to overlay your temperature record onto the other records. They can be compared simply by looking at the differences in their trends.

Re: This figure is revised according to your suggestion (Fig. 2).

Extended Data Figure 4: Please choose a colorblind figure color palette for panel B.

Re: The color palette for panel B is revised (Supplementary Fig. 10).

Extended Data Figure 7: Wouldn't the topical calibration of Zhao et al. (2023) be more appropriate here? The global calibration is intended for deep time applications.

Re: The tropical multivariate linear regression calibration of Zhao et al. (2023) is used (Supplementary Fig. 13).

References

- Allen, M. R. & Ingram, W. J. Constraints on future changes in climate and the hydrologic cycle. *Nature* **419**, 224–232 (2002).
- An, Z. et al. Glacial-interglacial Indian summer monsoon dynamics. *Science* **333**, 719–723 (2011).
- Baxter, A. J., Hopmans, E. C., Russell, J. M. & Sinninghe Damsté, J. S. Bacterial GMGTs in East African lake sediments: Their potential as palaeotemperature indicators. *Geochim. Cosmochim. Acta* **259**, 155–169 (2019).

- Baxter, A. J. et al. Disentangling influences of climate variability and lake-system evolution on climate proxies derived from isoprenoid and branched glycerol dialkyl glycerol tetraethers (GDGTs): the 250 kyr Lake Chala record. *Biogeosciences* **21**, 2877–2908 (2024).
- Bлага, C. I., Reichert, G. -J., Heiri, O. & Sinninghe Damsté, J. S. Tetraether membrane lipid distributions in water-column particulate matter and sediments: a study of 47 European lakes along a north-south transect. *J. Paleolimnol.* **41**, 523–540 (2009).
- Brigham-Grette, J. et al. Pliocene Warmth, Polar Amplification, and Stepped Pleistocene Cooling Recorded in NE Arctic Russia. *Science* **340**, 1421–1427 (2013).
- Clark, P. U., Alley, R. B. & Pollard, D. Northern Hemisphere Ice-Sheet Influences on Global Climate Change. *Science* **286**, 1104–1111 (1999).
- Clark, P. U., Shakun, J. D., Rosenthal, Y., Köhler, P. & Bartlein, P. J. Global and regional temperature change over the past 4.5 million years. *Science* **383**, 884–890 (2024).
- Daniels, W. C. et al. Archaeal lipids reveal climate-driven changes in microbial ecology at Lake El'gygytgyn (Far East Russia) during the Plio-Pleistocene. *J. Quat. Sci.* **37**, 900–914 (2022).
- Johnson, T. C. et al. A progressively wetter climate in southern East Africa over the past 1.3 million years. *Nature* **537**, 220–224 (2016).
- Mackay, A. W. The paleoclimatology of Lake Baikal: A diatom synthesis and prospectus. *Earth Sci. Rev.* **82**, 181–215 (2007).
- Martínez-Sosa, P. et al. A global Bayesian temperature calibration for lacustrine brGDGTs. *Geochim. Cosmochim. Acta* **305**, 87–105 (2021).
- Melles, M. et al. 2.8 Million Years of Arctic Climate Change from Lake El' gygytgyn, NE Russia. *Science* **337**, 315–320 (2012).
- Novak, J. B. et al. The Branched GDGT Isomer Ratio Refines Lacustrine Paleotemperature Estimates. *Geochem. Geophys. Geosy.* **26**, e2024GC012069 (2025).
- Peterse, F. et al. Decoupled warming and monsoon precipitation in East Asia over the last deglaciation. *Earth Planet. Sci. Lett.* **301**, 256–264 (2011).
- Prokopenko, A. A. et al. The link between tectonic and paleoclimatic events at 2.8–2.5Ma BP in the Lake Baikal region. *Quat. Int.* **80–81**, 37–46 (2001).
- Prokopenko, A. A., Hinnov, L. A., Williams, D. F. & Kuzmin, M. I. Orbital forcing of continental climate during the Pleistocene: a complete astronomically tuned climatic record from Lake Baikal, SE Siberia. *Quat. Sci. Rev.* **25**, 3431–3457 (2006).
- Qiang, X., Xu, X., Zhao, H. & Fu, C. Greigite formed in early Pleistocene lacustrine sediments from the Heqing Basin, southwest China, and its paleoenvironmental implications. *J. Asian Earth Sci.* **156**, 256–264 (2018).
- Tierney, J. E. et al. Environmental controls on branched tetraether lipid distributions in tropical East African lake sediments. *Geochim. Cosmochim. Acta* **74**, 4902–4918

(2010).

- Tierney, J. E. et al. Late-twentieth-century warming in Lake Tanganyika unprecedented since AD 500. *Nat. Geosci.* **3**, 422–425 (2010).
- Wang, B., Liu, J., Kim, H. -J., Webster, P. J., & Yim, S.-Y. Recent change of the global monsoon precipitation (1979 – 2008). *Clim. Dyn.* **39**(5), 1123–1135 (2011).
- Wang, H., et al. Salinity-controlled isomerization of lacustrine brGDGTs impacts the associated MBT'_{5ME} terrestrial temperature index. *Geochim. Cosmochim. Acta* **305**, 33–48 (2021).
- Wang, H. et al. New calibration of terrestrial brGDGT paleothermometer deconvolves distinct temperature responses of two isomer sets. *Earth Planet. Sci. Lett.* **626**, 118497 (2024).
- Xiao, W. et al. Ubiquitous production of branched glycerol dialkyl glycerol tetraethers (brGDGTs) in global marine environments: a new source indicator for brGDGTs. *Biogeosciences* **13**, 5883–5894 (2016).
- Yang, X., Jin, Z., Zhang, F. & Ma, X. Glacial-interglacial lake hydrochemistry in step with the Pleistocene Indian summer monsoon at the southeastern Tibetan Plateau. *Quat. Sci. Rev.* **329**, 108556 (2024).
- Zhao, B. et al. Evaluating global temperature calibrations for lacustrine branched GDGTs: Seasonal variability, paleoclimate implications, and future directions. *Quat. Sci. Rev.* **310**, 108124 (2023).
- Zhao, C. et al. Possible obliquity-forced warmth in southern Asia during the last glacial stage. *Sci. Bull.* **66**, 1136–1145 (2021).
- Zhao, Y., et al. Temperature reconstructions for the last 1.74-Ma on the eastern Tibetan Plateau based on a novel pollen-based quantitative method. *Global Planet. Change* **199**, 103433 (2021).

Point-by-point response to the reviewers' comments

Reviewer #1 (Remarks to the Author):

Dear authors,

after reading the revised version of your manuscript, I can conclude that you have addressed all of my comments and revised the manuscript accordingly. I have no further concerns or questions.

Yours sincerely,

Jiri Laurin

Re: Dear Jiri Laurin, thanks again for reviewing our revised manuscript and we are glad to know that you are satisfied with our revision.

Reviewer #2 (Remarks to the Author):

I appreciate the authors answering my concerns. However, I don't think the main concerns of how the data is presented was solved and I think the message of the manuscript needs a revision after the reviews that have already been done. Additionally, there is no discussion of how the high elevation (>1000m) of the archive can affect the temperature seen in the record, which is fundamental for interpretations of land temperature reconstructions.

Re: We greatly appreciate your comments on our revised manuscript. We have studied your comments carefully and made corrections or explanations accordingly. Briefly, we have provided new statistical results for data uncertainties, deleted unnecessary discussion on other terrestrial records, and added the discussion of the effect of elevation on long-term temperature trends. We are grateful for those comments and suggestions that further improved the quality of our work. Please see below for our point-by-point responses.

Even though you think the non-thermal error is small because of the lithology of the core you don't have a way to quantitatively address it. I think what you need is an appropriate propagation of error in your study to report your warming temperatures, and then evaluate if it is statistically significant. You NEED to incorporate the RMSE of the calibration you are using.

Re: Thanks for this comment. We acknowledge that the non-thermal influences cannot be exactly quantified although they are small (or negligible) in our record. To address your concern, we conservatively compounded the calibration error with analytical error to calculate the total error for our temperature reconstruction, according to Tierney et al. (2010, *Nat. Geosci.*) (Lines 379-383). The ± 1.8 °C uncertainty for absolute temperature values was visualized in Fig. 2a.

We should point out that while absolute temperature estimates should be interpreted with caution, relative temperature changes are believed to be reliable for GDGT-based reconstructions, as it can reduce systematic calibration error and site-specific effects (Tierney et al., 2010, *Nat. Geosci.*; Peterse et al., 2011, *Earth Planet. Sci. Lett.*; Johnson et al., 2016, *Nature*; Loomis et al., 2017, *Sci. Adv.*). Monte Carlo uncertainty

propagation has the advantage of both being easy to interpret and allowing for a wide variety of uncertainty distributions. Therefore, for our HQ temperature record (1.8–0.6 Ma), we further performed Monte Carlo simulations to calculate the uncertainty for the linear trend of temperature variation during 1.8–0.6 Ma, under the 1.8 °C error of absolute temperature values (Lines 383-392). We found that 1000 times of Monte Carlo simulations incorporating the 1.8 °C uncertainty all show warming trends, with an average value of 1.0 ± 0.3 °C/Myr (95% confidence interval = 0.4–1.7) (Fig. 3a, Supplementary Fig. 9), and the proportion of significant ($p < 0.05$) regressions is 78% (Lines 181-185). This indicates that the long-term warming trend at the Heqing Basin is statistically robust.

You mentioned a “long term” warming in your abstract, however, there is no quantitative result of the warming or any temperature reconstruction. I encourage you to add a quantitative result with the appropriate propagation error to make your abstract clearer and easier for readers to assess the impact of the study.

Re: Thanks for this suggestion and accordingly, we have added the quantitative result (1.0 ± 0.3 °C/million years) in the abstract (Line 25). The propagation error is 0.3 °C based on 1000 times of Monte Carlo simulations incorporating the 1.8 °C uncertainty of quantitative temperature reconstruction.

And in response to your reply of my first comment, I am glad that you show the monte carlo simulation of the slope, but it still doesn't show the appropriate error propagation of the brGDGT-based temperature reconstruction. I want to see the appropriate lower and upper boundary of your results.

Re: Based on Monte Carlo Uncertainty Analysis, the propagation error is 0.3 °C, and the lower and upper boundary is 0.4 and 1.7 °C/Myr, respectively at 95% confidence level. The 95% confidence interval for the warming trend from 1.8 to 0.6 Ma was visualized in Fig. 3a and Supplementary Fig. 9. This was added in Lines 181-184.

Lake Baikal: seems to me that after answering my concern this is not a useful record to show in your study. Don't show data that doesn't support your hypothesis or the main point of the paper. It's a distraction. It is not improper to not include data in your figures that doesn't help with the story you are telling. You are more than welcome to mention them and include them in suppl. Material but it distracts in short format papers.

Lake Malawi: 2.9°C/Myr, please add the uncertainties and appropriate error propagation for this. And same comment as Lake Baikal.

Re: Thanks for these suggestions. The Lake Baikal and Lake Malawi records were removed from Fig. 3, and the discussion on the two records were moved to Supplementary Information (Supplementary Discussion). The uncertainty (1.5 °C/Myr) and 95% confidence interval (0.2–5.9 °C/Myr) for the warming trend at Lake Malawi was also added in Supplementary Discussion.

Now that you clarify the elevation and included such data... What about changes in lapse rate? How this high-elevation (above 1000m) record is going to amplify the regional temperature trend? Is the warming that you are seeing an amplification of temperature due to elevation and not a response of the mechanism? I think you need to expand more on this and explore the elevation-dependent warming or amplification/changes in temperature. I suggest you do it in a way that you can differentiate if you 1°C is a regional warming or a temperature amplification due to elevation.

Re: Thanks for this suggestion. We fully agree with you that warming at the Heqing Basin might be amplified due to elevation-dependent warming, but this effect cannot invert the temperature trends and thus can hardly explain the warming instead of cooling trend of Pleistocene terrestrial temperature. Following your suggestion, we added a paragraph specifically discussing the potential effect of elevation on the warming trend in East Asia in Lines 229-243 as “The long-term warming trend during 1.8–0.6 Ma observed for the 3 East Asian records at Heqing, Zoige and Lingtai, in contrast to the contemporaneous global sea surface cooling, cannot be primarily attributed to the relatively high elevation of these terrestrial records. Observational data and numerical models for global temperature variations suggest that, elevated land surfaces warm faster than non-elevated ones during the past several decades⁴⁶. On the other hand, GDGT-based temperature reconstructions from East African lakes demonstrate that cooling during the Last Glacial Maximum was amplified with elevation, resulting in a significantly steeper lapse rate than today⁴⁷. However, while elevation might amplify the amplitude of temperature variations, the warming or cooling trends at different elevations cannot be inverted. Based on ERA5 reanalysis, HadCRUT5 dataset, and CMIP6 historical simulations (1959–2014), land surface warming is on average 16% higher at 2190 m a.s.l than that at sea level across the tropics and subtropics (40° S to 40° N)⁴⁶. Assuming a similar elevation-dependent warming at Heqing, the warming trend should be 0.9 °C/Myr at sea level, still opposite to the 2.5 °C/Myr cooling for global SST.”

Lines 164-171: please show such ratios in a supplementary figure for all samples to assess how different they are. Please add the references for which those ratios are used to exclude GDGTs that are influenced by “non-thermal” factors.

Re: Thanks for this suggestion. GDGT-based indices, including IIIa/IIa (Supplementary Fig. 2c) and GDGT-0/cren (Supplementary Fig. 2d), were provided in Supplementary Fig. 2. The references for which those ratios are used to constrain non-thermal influences on the brGDGT paleothermometer were added in Line 152.

A suggestion for your Fig. S12. use a red star without filling, not a solid point, so readers can look at the color at your site. Interesting that you note that that map is consistent with your data but seems inconsistent with other terrestrial data that you showed.

Also, your Figure R3. works as a better data visualization for what you want to say... maybe consider including it in the suppl.

Re: Thanks. Following your suggestion, the solid point was replaced by a red star with filling in **a** (now for Supplementary Fig. 14), and Supplementary Fig. 14c&d were replaced by Fig. R3. Moreover, we have deleted the discussion and curves of the Lake Baikal, Lake El'gygytgyn, and Lake Malawi records in the main text, and the remaining 3 records from East Asia are consistent with modeling output.

I suggest revisiting the figures you are showing in the main text and focus on the data that helps you support your argument (or main message of the paper). Right now for me looks confusing.

Re: Thanks. The Lake Baikal and Lake Malawi records were removed from Fig. 3, and linear regressions with 95% confidence intervals were added. Moreover, the uncertainty of the temperature calibration (± 1.8 °C) was presented more clearly in Fig. 2a.

Under data availability: what is figshare? Where is the data going to be stored for open access? Make sure you include each individual GDGT (fractional abundance, branched

and iso) on it.

Re: Figshare is a web-based interface designed for academic research data management and research data dissemination. Springer Nature is partnering with figshare at a number of Nature Portfolio and Academic Journals, offering a straightforward repository option for a wide range of data types. The URL for our data was added in Line 396 and will become public after the paper is published (the Figshare private link is <https://figshare.com/s/7867781e5620cf96c5e0>). It consists of all related GDGT information for this work, including the fractional abundances of individual brGDGTs, GDGT proxies including GDGT-0/cren ratio for isoGDGTs, and brGDGT concentration.

Reviewer #4 (Remarks to the Author):

Wang et al. presents a 2-million-year temperature records from Heqing Paleolake in southwestern China. The main findings include a warming trend, instead of cooling, during the time interval of 1.8-0.6 Ma. The authors link this to the growth of the Antarctic Ice Sheet. The revision is well-written, and I enjoy reading this story.

I joined the peer review process after the first round and mainly evaluated the revision/supplements and the rebuttal letter, especially the communications between the authors and reviewer #3. The main concern centers on the insufficient assessment of the non-thermal effects on the brGDGTs, thus will undermine the use of brGDGT as a temperature indicator in this study. After carefully reading the revision, I feel the authors have largely incorporated the Reviewer's comments and have added a thorough analysis to properly discuss the potential non-thermal influence. I do believe a rigorous assessment of the brGDGT is a must for such an important and long terrestrial temperature record, and I appreciate the authors' efforts to address those questions.

Re: Many thanks for your positive assessment of our work.

I want to echo one important concern that was raised by Reviewer 3 regarding the calibration used in this study. The authors have done a good job of arguing that the brGDGTs are mainly influenced by temperatures, and thus can be a good temperature indicator. What is missing (in the paragraphs around line 125) here is to justify that the best calibration is the one from Wang et al. (2024) EPSL. I encourage the authors to add some discussions here, and I believe it's not hard, given that the Supplementary Figure 13 shows very similar patterns using different calibrations. The authors briefly touch on this in the method section, but I feel some discussion in the main text is essential. So far, the authors have spent quite a lot of time ruling out the non-thermal effects in the first subsection, and this is the time to directly add information that brGDGTs can be used to reconstruct temperature successfully with this calibration. A few sentences stating the advantages of this calibration and why it fits the best will be very helpful.

Re: Thanks for your suggestion. In the revised manuscript Lines 141-150), we further explained why we applied the calibration of Wang et al. (2024, *Earth Planet. Sci. Lett.*) after ruling out the significant effects of non-thermal factors in the first subsection as "The applications of different lacustrine calibrations based on various statistical methods^{11,14,29-31} show similar trends and amplitudes of MAT reconstructions, although the absolute values might differ (Supplementary Fig. 7). Nevertheless, we note that reconstructed MAT based on 4 calibrations^{11,14,30,31} is weakly but significantly

correlated with IR_{6ME} ($p \leq 0.05$) (Supplementary Fig. 3e-h), pointing to a potentially minor impact of brGDGT isomerization on these calibrations. On the other hand, reconstructed temperature based on the recent calibration²⁹ which may mitigate the isomer effect has no correlation with IR_{6ME} ($r = 0.08$, $p = 0.14$) throughout the core (Supplementary Fig. 3d), and therefore, this calibration was applied for quantitative temperature reconstruction”.

The figures showing soil GDGTs are missing in the supplements. The authors call out Supplementary Fig. 2 in Lines 84, 88, and 92, but that is not the correct one.

Re: Thanks for this comment. In the previous version of Supplementary Fig. 2, we used blue dashed lines in a-c to indicate averaged proxy values for surrounding soils. Following your comment, we further added Box and Whisker Plots for soil GDGTs in the revised Supplementary Fig. 2a-c.

Line 101, regarding the GDGT-0/cren, please consider citing Schneider et al. (2024) QSR. DOI: <https://doi.org/10.1016/j.quascirev.2024.108851>

Re: Thanks, and this reference (Schneider et al., 2024, *Quat. Sci. Rev.*) was added in our revised manuscript (Line 99 and Lines 462-464).

References

- Baxter, A. J. et al. Disentangling influences of climate variability and lake-system evolution on climate proxies derived from isoprenoid and branched glycerol dialkyl glycerol tetraethers (GDGTs): the 250 kyr Lake Chala record. *Biogeosciences* **21**, 2877–2908 (2024).
- Bлага, C. I., Reichert, G. -J., Heiri, O. & Sinninghe Damsté, J. S. Tetraether membrane lipid distributions in water-column particulate matter and sediments: a study of 47 European lakes along a north-south transect. *J. Paleolimnol.* **41**, 523–540 (2009).
- Byrne, M. P., Boos, W. R., Hu, S. Elevation-dependent warming: observations, models, and energetic mechanisms. *Weather Clim. Dynam.* **5**, 763–777 (2024).
- Daniels, W. C. et al. Archaeal lipids reveal climate-driven changes in microbial ecology at Lake El'gygytgyn (Far East Russia) during the Plio-Pleistocene. *J. Quat. Sci.* **37**, 900–914 (2022).
- Johnson, T. C. et al. A progressively wetter climate in southern East Africa over the past 1.3 million years. *Nature* **537**, 220–224 (2016).
- Loomis, S.E. et al. The tropical lapse rate steepened during the Last Glacial Maximum. *Sci. Adv.* **3**, e1600815 (2017).
- Peterse, F. et al. Decoupled warming and monsoon precipitation in East Asia over the last deglaciation. *Earth Planet. Sci. Lett.* **301**, 256–264 (2011).
- Schneider, T. et al. Tracing Holocene temperatures and human impact in a Greenlandic Lake: Novel insights from hyperspectral imaging and lipid biomarkers. *Quat. Sci. Rev.* **339**, 108851 (2024).
- Tierney, J. E. et al. Late-twentieth-century warming in Lake Tanganyika unprecedented since AD 500. *Nat. Geosci.* **3**, 422–425 (2010).

- Wang, H. et al. New calibration of terrestrial brGDGT paleothermometer deconvolves distinct temperature responses of two isomer sets. *Earth Planet. Sci. Lett.* **626**, 118497 (2024).
- Xiao, W. et al. Ubiquitous production of branched glycerol dialkyl glycerol tetraethers (brGDGTs) in global marine environments: a new source indicator for brGDGTs. *Biogeosciences* **13**, 5883–5894 (2016).

Point-by-point response to the reviewers' comments

Reviewer #2 (Remarks to the Author):

I thank the authors for answering all my concerns. I think the paper is now ready for publication with only some minor comments that they will need to add before acceptance.

Re: Thanks again for reviewing our revised manuscript and we are pleased to know that you are satisfied with our revision. We are grateful for those comments and suggestions that further improved the quality of our work. Please see below for our point-by-point responses to your minor comments.

Please add the 95% confidence interval result in your abstract (see Tierney et al., 2025 AGUAdvances as an example of how to report it).

Re: Thanks for this suggestion and the 95% CI is added in Line 36.

The first sentence of your abstract should be something that highlights the importance of the period you are reconstructing... not just saying that we don't know terrestrial MAT. Why is it important to reconstruct terrestrial temperature in the Pleistocene? (again, you can look at the previous paper as an example)

Re: Thanks. We have added “limiting our understanding of the patterns, mechanisms and impacts of past temperature changes” after “How terrestrial mean annual temperature (MAT) evolved throughout the past 2 million years (Myr) remains elusive” according to your suggestion (Lines 31-32).

Line 35, replace with “Earth’s climate shifted from 41...”

Re: Accepted and revised (Line 46).

Line 233: non-elevated = than those near sea-level

Re: Thanks, and “non-elevated” was replaced by “those near sea-level” in Line 244.

Line 234: it makes no sense to start that sentence with: “on the other hand” when you are talking about another period of time, maybe you can say: “During the Last glacial maximum, GDGT-based temperature reconstructions from East African lakes show amplified cooling with elevation...” both modern and LGM show that at higher elevation temperature amplification is bigger than near sea level.

Re: Thanks for this suggestion. Accordingly, we have revised this part as “During the Last glacial maximum (LGM), GDGT-based temperature reconstructions from East African lakes show amplified cooling with elevation, resulting in a significantly steeper lapse rate than today. Both modern and LGM data show that at higher elevation temperature amplification is greater than near sea level ...” in Lines 245-249.

about the data availability: the link that was provided to me does not show the fractional abundances of branched and iso GDGT or neither the ratios calculated in this study. It only shows the Age (Ma), MAT, and MAT-error. PLEASE make sure to upload all of the data so your study is reproducible.

Re: Thanks for emphasizing this point. The URL for our data is provided in DATA

AVAILABILITY. It consists of all related GDGT information for this work, including the fractional abundances of individual brGDGTs, GDGT proxies including GDGT-0/cren ratio for isoGDGTs, and brGDGT concentration.

Reviewer #4 (Remarks to the Author):

My concerns have been resolved after reading the revision and the response letter.

Re: We are pleased to know that you are satisfied with our revision.